

# A comparison between parameter regionalization and model calibration with flow duration curves for prediction in ungauged catchments

Daeha Kim[1], Ilwon Jung[2], Jong Ahn Chun[1]

[1]APEC Climate Center, Busan, 48058, South Korea
[2]Korea Infrastructure Safety & Technology Corporation, Jinju, Gyeongsangnam-do, 52852, South Korea

*Correspondence to*: Daeha Kim (d.kim@apcc21.org)

**Abstract.** Streamflow prediction in ungauged catchments is essential for hydrological applications because most catchments in the world are ungauged. The flow duration curve (FDC) has received increasing attention as a criterion for calibration of

runoff models in ungauged catchments; however, performance of the FDC-based model calibration was barely evaluated in comparison with parameter regionalization. In this study, we conducted a comparative assessment between the conventional proximity-based parameter regionalization and the model calibration with regional FDCs. Leave-one-out evaluations of the two methods showed that the proximity-based parameter regionalization outperformed the model calibration with regionalized FDCs in hydrograph prediction. No information of flow timing in FDCs seems to cause high uncertainty in flow

predictions. The relative merits of the model calibration with regional FDCs were low bias in predicted flows and high reproducibility of the baseflow index. From the evaluation of flow signature reproducibility, it was indicated that the rising limb density is likely to be orthogonal to the FDCs, and thus additional conditioning with the rising limb density is expected to improve the model calibration with regional FDCs.

## 1 Introduction

Streamflow is essential information for hydrological applications such as calibration of rainfall-runoff models, water resources planning and management, disaster risk management, and environmental impact assessment (Westerberg et al., 2014; Parajka et al., 2013); however, most catchments in the world are remained as ungauged. Even in regions with dense river gauging networks, streamflow observations are available only at river sections where water stages are monitored (Viglione et al., 2013). Most catchments of interest have no streamflow data for hydrologic applications, thus tools for

prediction in ungauged catchments are inevitably required.

A standard method for hydrograph prediction in ungauged catchment is the regionalization of model parameters. Hydrological models that simulate the time series of streamflow are typically calibrated with observed hydrographs (e.g., Oudin et al., 2006; Gupta et al., 2009). The parameter regionalization is to transfer the calibrated parameters from gauged to ungauged catchments based on geographical proximity, functional similarity, and regression with catchment characteristics



(e.g., Seibert and Beven, 2009; Yadav et al., 2007; Parajka et al., 2007; Wagener and Wheater, 2006; Dunn and Lilly, 2001). The parameter regionalization has provided convenience in prediction of hydrographs with a long tradition (Wagener et al., 2004); nevertheless, previous studies have cast concerns about errors in input data and structural deficiency of models as well as over-reliance on calibration in gauged sites (Hrachowitz et al., 2013). In addition, it has been a particular concern that

the calibrated parameters can have weak uniqueness for capturing functional behaviour of catchments (Bárdossy, 2007; Oudin et al., 2008) due to the equifinality, and eventually it leads to unreliable parameter regionalization.

An alternative to the parameter regionalization is the use of measurable physical properties or regionalized flow signatures for direct estimation of model parameters for ungauged catchments. Fang et al. (2010), for instance, found that a priori parameters derived from remotely-sensed data could provide substantial predictive performance. The flow signatures, i.e.,

metrics or auxiliary data representing catchment behaviour, also have been frequently applied for model calibration. Examples includes the use of isotope concentrations (e.g., Son and Sivapalan, 2007), the baseflow index (Bulygina et al., 2009), the spectral density of streamflow (Montanari and Toth, 2007; Winsemius et al., 2009), and among others.

In particular, the flow duration curve (FDC) has received great attention as a single criterion for model calibration (e.g., Westerberg et al., 2011; 2014; Yu and Yang, 2000). An empirical FDC, i.e., the relationship between flow magnitude and its

exceedance probability, represents the flow regime of a given catchments, and thus is regarded as a key signature of the catchment behaviour (Pugliese et al., 2014; Yokoo and Sivapalan, 2011). Many efforts have been made for transferring FDCs from gauged to ungauged catchments, and, in turn, FDC regionalization achieved high predictive performance (e.g., Mohamoud, 2008; Pugliese et al., 2014). Furthermore, recent in-depth studies shed light on physical controls of spatial patterns of empirical FDCs (Yaeger et al., 2012; Ye et al., 2012). Given the improved skills in FDC predictions, the use of

regionalized FDCs for model calibration may provide better hydrograph predictions in ungauged catchments because it can circumvent the concerns in parameter regionalization.

However, several questions arise when using regionalized FDCs for calibrating model parameters. Firstly, FDCs have information of flow magnitudes only; thereby the parameter sets fitted to FDCs may show deficiency in predicting flow timing (van Werkhoven et al., 2009). Indeed, regionalized FDCs also have uncertainty introduced by streamflow data and

structural errors in the regionalization methods. It is uncertain if model parameters calibrated to a regionalized FDC would have less predictive errors than regionalized parameters. Comparative assessments are still uncommon on FDC-based calibrations while parameter regionalization has been comprehensively evaluated (e.g., Parajka et al., 2013; Oudin et al., 2008). Secondly, there may be other flow signatures orthogonal to FDCs. If additional signatures condition the FDC-based calibration, the calibrated parameters will perform better because of alleviated equifinality; nevertheless, it is still an open

question which flow signatures complement FDCs.

The objective of this study, therefore, is to evaluate predictive performance of model calibration with regionalized FDCs in comparison with the conventional parameter regionalization. We compared a proximity-based parameter regionalization and the model calibration with regionalized FDCs. The two methods are evaluated in terms of hydrograph and FDC





predictability by leave-one-out cross-evaluations. We also assessed the two methods in terms of reproducibility of major flow signatures so as to find flow signatures orthogonal to regional FDCs.

## 2 Description of the study area and data

The study area is 45 gauged catchments located across South Korea with no or negligible human-made alteration (e.g., river

diversion and dam operations) in flow regime (Fig. 1). South Korea is characterized as a temperate and humid climate with rainy summer seasons. The North Pacific high-pressure brings monsoon rainfall with high temperatures in summer seasons, while dry and cold weathers prevail in winter seasons due to the Siberian high-pressure. A typical range of annual precipitation is 1,000-1,900 mm, and approximately 70 percent of precipitation falls in summer seasons (from June to September). Streamflow usually peaks in the middle of a summer season because of heavy rainfall or typhoons, and hence

information of catchment behaviour is largely concentrated in summer-season hydrographs. Snow accumulation and ablation are observed at high elevations, but their effects on flow regime are minor due to the limited amount of winter precipitation.

The gauged catchments in Fig. 1 were selected based on availability of streamflow data. Although long streamflow data are available at a few river gauging stations, high-quality streamflow data across the entire South Korea have been produced since establishment of the Hydrological Survey Center in 2007. Hence, we collected streamflow data at 29 river gauging

stations from 2007 to 2015 as well as inflow data operationally recorded at 16 multi-purpose dams from the Water Resources Management Information System operated by the Mistry of Land, Infrastructure, and Transport of the Korean government (available at http://www.wamis.go.kr/). Streamflow data from 2011 to 2015 were used for calibrating parameters of GR4J model and regionalizing empirical FDCs, while those from 2007 to 2010 were separately taken for checking validity of the estimated parameters and leave-one-out cross-evaluations. The catchment information is summarized in Table 1.

As the climatic inputs for rainfall-runoff modelling, we used gridded daily precipitation, and maximum and minimum temperatures at 3-km grid resolution produced by spatial interpolation between 60 climatic stations of the Korean Meteorological Administration. Jung and Eum (2015) combined the Parameter-elevation Regression on Independent Slope Model (Daly et al., 2008) and the inverse distance method for the spatial interpolation, and found improved performance for producing gridded precipitation and temperatures in South Korea. The enhanced gridded climatic data from 1973 to 2015 are

internally available at the APEC Climate Center (will be distributed via the APCC Data Service System at http://adss.apcc21.org/), and were used for modelling hydrographs at outlets of the catchments. Processing the climatic data for GR4J model will appear later in the methodology section.





## 3 Methodology

### 3.1 Calibration of GR4J model using observed hydrographs

A conceptual rainfall-runoff model, GR4J (Perrin et al., 2003), was selected for simulating hydrographs of the gauged catchments, and is schematized in Figure 2. GR4J conceptualizes catchment response to rainfall with four free parameters

that regulate the water balance and water transfer functions. Since its parsimonious and efficient structure enables robust calibration and reliable regionalization of model parameters, GR4J has been frequently used for modelling daily hydrographs with various purposes (e.g., Nepal et al, 2016; Tian et al., 2013). It is classified as a soil moisture accounting model, and more details are found in Perrin et al. (2003).

Before calibration of GR4J in the gauged catchments, we preliminarily processed the gridded climatic data to convert

precipitation data to liquid water depths forcing catchments (i.e., rainfall and snowmelt depths) using a simple physics-based snowmelt model proposed by Walter et al. (2005). The snowmelt model has the same input requirement as GR4J, thus no additional data are necessary for the processing. It simulates point-scale snow accumulation and ablation processes based on the energy balance, and produces the liquid water depths and snow water equivalent as outputs. The preliminary processing makes it possible to consider natural water holding capacity of snowpack, which is not included in GR4J, with no additional

parameters. Though combining a temperature index snowmelt model with GR4J can be an alternative approach, it increases the number of parameters and thus model uncertainty. Since contribution of snowmelt to the entire flow regime is insignificant in South Korea as described, maintaining the parsimonious structure of GR4J would be more important for parameter calibration and regionalization albeit the physical snowmelt model also has error sources. The preliminary processing was for reducing systematic errors from no snow component in GR4J. After the snowmelt modelling, we

spatially averaged pixel values of the liquid water depths and maximum and minimum temperatures within the boundary of each catchment as lumped climatic inputs for GR4J.

Besides, we checked consistency between the spatially-averaged liquid water depths and observed hydrographs using the current precipitation index (CPI; Smakhtin and Masse, 2000) defined as:

$$I_t = I_{t-1} \cdot K + R_t \tag{1}$$

where $I_t$ is the CPI (mm) at day t, K is a decay coefficient (0.85 d$^{-1}$), and $R_t$ is the liquid water depth (mm d$^{-1}$) at day t that forces the catchment (i.e., rainfall or snowmelt). CPI mimics temporal variations in typical streamflow data by converting intermittent rainfall events to a continuous time series with an assumption of the linear reservoir. The correlation between CPI and observed flows can be conveniently used to check consistency between model input and output when calibrating parameters of runoff models (e.g., Westerberg et al., 2014).

The correlation coefficients between CPI and observed flows in the 45 catchments had an average of 0.67 with a range of 0.43-0.79. No outliers were found in the box plot of the correlation coefficients. Hence, we hypothesized that acceptable consistency existed between model inputs and observed flows.





After the consistency check, we calibrated GR4J parameters using the shuffled complex evolution (SCE) algorithm (Duan et al., 1992) that has been frequently adopted (e.g., Cooper et al., 2007) due to its robustness and fast convergence. Based on convenience and normalized measure of model performance, we chose a traditional objective function to maximize the Nash-Sutcliffe efficiency ($NSE_Q$) between observed flows ($Q_{obs}$) and simulated flows ($Q_{sim}$) as:

$$NSE_Q = 1 - \frac{\sum_{t=1}^{T}(Q_{obs,t} - Q_{sim,t})^2}{\sum_{t=1}^{T}(Q_{obs,t} - \overline{Q_{obs}})^2} \tag{2}$$

where t and T are the observed time and the length of $Q_{obs}$ respectively, and $\overline{Q_{obs}}$ is the temporal average of $Q_{obs}$. The optimal parameter set of each catchment was determined as one that maximizes $NSE_Q$ for the calibration period (2011-2015). The hydrograph simulated with the optimized set was compared to observed hydrograph during the validation period (2007-2010). We had a two-year warm-up period for initializing all runoff simulations.

## 3.2 Proximity-based regionalization of flow duration curves

We adopted a state-of-the-art geostatistical method proposed by Pugliese et al. (2014) for regionalizing empirical FDCs of the gauged catchments. Pugliese et al. (2014) used the top-kriging method (Skøien et al., 2006) to spatially interpolate the total negative deviation (TND), which indicates an area between the mean annual flow and below-mean flows in a normalized FDC. The Top-kriging weights that interpolate TND values were used as weights to estimate flow quantiles of ungauged catchments from empirical FDCs of neighbouring gauged catchments. Since the top-kriging weights are obtained based on topological proximity between catchments, the geostatistical method is a proximity-based regionalization of FDCs. The FDC of an ungauged catchment is estimated from normalized FDCs of neighbouring gauged catchments as:

$$\widehat{\Phi}(w_0, p) = \widehat{\phi}(w_0, p) \cdot \overline{Q}(w_0) \tag{3a}$$

$$\widehat{\phi}(w_0, p) = \sum_{i=1}^{n} \lambda_i \cdot \phi_i(w_i, p), \quad p\epsilon(0,1) \tag{3b}$$

where $\widehat{\Phi}(w_0, p)$ is the estimated quantile flow ($m^3 s^{-1}$) at an exceedance probability p (unitless) for an ungauged catchment $w_0$, $\widehat{\phi}(w_0, p)$ is the estimated normalized quantile flow (unitless), $\overline{Q}(w_0)$ is the annual mean streamflow ($m^3 s^{-1}$) of the ungauged catchment, and $\phi_i(w_i, p)$ and $\lambda_i$ are normalized quantile flows (unitless) and corresponding top-kriging weights (unitless) of gauged catchment $w_i$ respectively. The unknown mean annual flow of a ungauged catchment, $\overline{Q}(w_0)$, can be estimated with a rescaled mean annual precipitation defined as:

$$MAP^* = 3.171 \times 10^{-5} \cdot MAP \cdot A \tag{4}$$

where MAP* is the rescaled mean annual precipitation ($m^3 s^{-1}$), MAP is mean annual precipitation ($mm yr^{-1}$) and A is drainage area ($km^2$) of the ungauged catchment. If MAP* is used for regionalizing FDCs, empirical FDCs of gauged catchments should be normalized by their MAP* values too.





A distinct advantage of the geostatistical method is that it can estimate the entire flow quantiles in a FDC with a single set of top-kriging weights. Since a parametric regional FDC (e.g., Yu et al., 2002; Mohamoud, 2008) is commonly obtained from independent models for estimating each flow quantile, e.g., multiple regressions between each quantile and catchment properties, fundamental characteristics in a FDC continuum can be entirely or partly lost. The geostatistical method, on the other hand, treats all flow quantiles as a single object; thereby, features in a FDC continuum are preserved. It showed improved performance to reproduce empirical FDCs only using topological proximity between catchments. Further details and discussion are available in Pugliese et al. (2014).

For regionalization of FDCs, we first calculated TND values of each gauged catchment with observed flows during the calibration period (2011-2015). Then, we spatially interpolated the TND values using the rtop R-package (Skøien et al., 2014), and applied the top-kriging weights to quantile flow estimation. Regionalized FDCs were estimated at 103 evaluation points of exceedance probabilities (p of 0.001, 0.005, 99 points between 0.01 and 0.99 at an interval of 0.01, 0.995, and 0.999).

### 3.3 Evaluation of runoff predictions in ungauged catchments

We tested two methods for predicting hydrographs of ungauged catchments using GR4J model: (a) proximity-based parameter transfer; and (b) parameter calibration using the regionalized FDC. The former is a typical method that transfers estimated parameters of nearby gauged catchments to a target ungauged catchment whereas the latter fits model parameters to a regionalized flow signature of the target ungauged catchment.

To evaluate hydrograph predictability of two methods, we used two performance indicators. One is $NSE_Q$ in Eq. (2), which indicates the proportion of the variance in the observed hydrograph that is predictable from the modelled hydrograph. The other is the volume error ($VE_Q$) that indicates the relative absolute bias between the modelled and observed hydrographs as:

$$VE_Q = \frac{\sum_{t=1}^{T}|Q_{sim,t} - Q_{obs,t}|}{\sum_{t=1}^{T} Q_{obs,t}} \tag{5}$$

To measure FDC predictability, we compared simulated FDCs with observed ones in terms of NSE between observed and simulated flow quantiles ($NSE_{FDC}$) defined as:

$$NSE_{FDC} = 1 - \frac{\sum_{p=1}^{P}(Q_{sim,p} - Q_{obs,p})^2}{\sum_{p=1}^{P}(Q_{obs,p} - \overline{Q_{obs,p}})^2} \tag{6}$$

where $Q_{sim,p}$ and $Q_{obs,p}$ are the estimated and observed flow quantiles at an exceedance probability p respectively, and $\overline{Q_{obs,p}}$ is the average of observed flow quantiles. The 103 points of the regional FDCs (i.e., P=103) were used for the cross-evaluations.

We additionally selected three flow signatures to evaluate flow signature predictability; the runoff ratio ($R_{QP}$), the baseflow index ($I_{BF}$), and the rising limb density ($R_{LD}$) defined as:





$$R_{QP} = \frac{\overline{Q}}{\overline{P}} \tag{7a}$$

$$I_{BF} = \sum_{t=1}^{T} \frac{Q_{B,t}}{Q_{obs,t}} \tag{7b}$$

$$R_{LD} = \frac{N_{RL}}{T_R} \tag{7c}$$

where $\overline{Q}$ and $\overline{P}$ are the average flow and precipitation during a period, $Q_{B,t}$ (m s$^{-1}$) is the base flow at time t, $N_{RL}$ is the number of rising limb, and $T_R$ is the total amount of time the hydrograph is rising (days). $Q_{B,t}$ is calculated by subtracting direct flow ($Q_{D,t}$) in m s$^{-1}$ from $Q_{obs,t}$ as:

$$Q_{D,t} = c \cdot Q_{D,t} + 0.5 \cdot (1 + c) \cdot (Q_{obs,t} - Q_{obs,t-1}) \tag{8a}$$

$$Q_{B,t} = Q_{obs,t} - Q_{D,t} \tag{8b}$$

where parameter c is a value of 0.925 from a comprehensive case study by Eckhardt (2007). Reproducibility of $R_{QP}$, $I_{BF}$, and $R_{LD}$ are evaluated by the relative absolute bias between modelled and observed signatures as:

$$D_{FS} = \frac{|FS_{sim} - FS_{obs}|}{FS_{obs}} \tag{9}$$

where $D_{FS}$ is the relative absolute bias, $FS_{sim}$ is a flow signature of the modelled flows, and $FS_{obs}$ is that of the observed flows. Details of the two regionalization methods are following.

### 3.3.1 Proximity-based parameter transfer from gauged to ungauged catchments

Regionalization of model parameters is a common approach for estimating hydrographs of ungauged catchments. It has three typical categories: (a) proximity-based parameter transfer (e.g., Oudin et al., 2008); (b) similarity-based parameter transfer (e.g., McIntyre et al., 2005); and (c) regression between parameters and physical properties of gauged catchments (e.g., Kim and Kaluarachchi, 2008). Predictive performance of the parameter regionalization depends on climatic conditions, complexity of the hydrologic model, among other factors (Parajka et al., 2013).

We chose the proximity-based parameter transfer for consistency with the proximity-based regionalization of empirical FDCs. The proximity-based parameter regionalization is often used because of its competitive performance and simplicity (Oudin et al., 2008; Parajka et al., 2013). We transferred calibrated parameter sets from five nearby catchments to a target catchment based on the findings in Oudin et al. (2008). The five simulations weighted by inverse distances between the target and donor catchments were averaged for representing the modelled hydrograph of a target ungauged catchment.

### 3.3.2 Parameter calibration using the regionalized FDCs

For comparison, the FDCs regionalized by the geostatistical method were directly used to calibrate GR4J model parameters for each catchment. Since flow signatures have been important constraints for parameter estimation in gauged catchments





(e.g., Pfannerstill et al., 2014; Westerberg et al., 2011), agreement between a regionalized FDC and one simulated by a rainfall-runoff model also can be the objective function for parameter calibration (e.g., Westerberg et al., 2014; Masih et al., 2010; Yu and Yang, 2000). The objective function of parameter calibration was to maximize $NSE_{FDC}$ between regionalized and simulated flow quantiles at the 103 exceedance probability (p) selected as the evaluation points.

A regionalized FDC is less informative than an observed hydrograph because of no information of flow timing and uncertainty in the geostatistical regionalization. A complete agreement between regionalized and modelled FDCs does not guarantee high predictive performance (van Werkhoven et al., 2009) in hydrograph prediction. Hence, the Monte Carlo simulations were used for maximization of $NSE_{FDC}$ instead of a complex optimization algorithm. We tested 10,000 parameter sets randomly generated within the given ranges of GR4J parameters (Table 2), and selected five sets for runoff

simulations by sorting corresponding $NSE_{FDC}$ values in descending order. The average flows of the five simulations weighted by corresponding $NSE_{FDC}$ values were taken as modelled flows of a target catchment.

## 4 Results

### 4.1 Calibration of GR4J parameters using observed hydrographs

The GR4J model was calibrated with observed hydrographs of the 45 catchments using the SCE algorithm, and Fig. 3a

shows $NSE_Q$ values for the calibration and validation periods. The averages of $NSE_Q$ values were 0.69 and 0.63 with ranges of 0.13-0.92 and 0.19-0.87 for the calibration and validation periods respectively. In spite of some losses of $NSE_Q$ from calibration to validation, GR4J was unlikely to be over-fitted and acceptably reproduced observed hydrographs in the case of relative high $NSE_Q$ values (>0.6). The predictive performance was highly dependent on consistency between the lumped input and streamflow as illustrated in Fig. 3b. The correlation coefficients CPI and observed flows have a clear positive

relationship with $NSE_Q$, and indicate that consistency between climatic inputs and observed flows is a prerequisite for high model performance.

Fig. 4 illustrates example observed and modelled hydrographs of two selected catchments. As mentioned earlier, catchment response to precipitation is highly concentrated on summer-season hydrographs. Whereas peaks and falling limbs of observed hydrographs were well captured by GR4J in the case of high $NSE_Q$ (Fig. 4a), some discrepancy between observed

and modelled hydrographs was found as $NSE_Q$ decreases (Fig. 4b). The less predictive performance can be attributed to lower consistency between model input and output, and/or structural deficiency of GR4J. The parsimonious structure of GR4J is good for conceptualizing general hydrological processes and regionalization for ungauged catchments, but can be an error source for catchments with uncommon features. The high $NSE_{FDC}$ values (> 0.95) in the both cases of Fig. 4 indicate that hydrograph reproducibility is a stricter measure for model calibration than predictability of flow frequency.

We assumed that the catchments with low $NSE_Q$ (<0.6) for the calibration period have unacceptable uncertainty either in model inputs or observed hydrographs. 34 catchments were of $NSE_Q$ higher than 0.6, and selected for the proximity-based





parameter regionalization (see Table 1). $NSE_Q$ values of the selected catchments have averages of 0.76 and 0.73 with ranges of 0.62-0.92 and 0.52-0.87 for the calibration and validation periods respectively.

## 4.2 Regionalization of FDCs using empirical FDCs

An area between the mean annual flow and below-mean flows (i.e., TND) in a FDC represents the catchment behaviour (Pugliese et al., 2014) such that the top-kriging weights for interpolating TND values can regionalize empirical FDCs. We followed the same procedure of Pugliese et al. (2014) for regionalizing empirical FDCs of the 45 catchments. We cross-validated TND values of the 45 catchments and obtained top-kriging weights ($\lambda_i$) using the rtop r-package. Then, the top-kriging weights were used to regionalize flow quantiles. The number of neighbours for the spatial interpolation was iteratively decided as n=5 at which additional neighbouring TNDs are unlikely to increase agreement between estimated and empirical TNDs.

Fig. 5a shows a 1:1 scatter plot between the empirical TND values and those spatially interpolated by the top-kriging method for the calibration period. The 45 catchments of this study recorded 0.56 of correlation coefficient between empirical and estimated TNDs (equivalent to 0.30 of NSE), while predicted TNDs of the 18 adjacent catchments of the original study (Pugliese et al., 2014) had a NSE of 0.6. The relatively low performance of top-kriging interpolation suggests that topological proximity is not the only factor explaining the spatial variation of TNDs. Some physical properties (e.g., catchment size, impervious area) can be significantly different even between two adjacent catchments; thereby large discrepancy can be caused between observed and estimated TNDs. Particularly, estimated TNDs of isolated catchments (e.g., catchments 3 and 5 in Fig. 1) can have errors from dissimilarity with the donor catchments. Spatial proximity is powerful for prediction in ungauged catchments (Merz and Blöshl, 2004), but imperfect to entail similarity in functional catchment behaviour (e.g., Ali et al., 2012).

Despite the abovementioned limitation, we found usefulness of the top-kriging weights for estimating flow quantiles. The $NSE_{FDC}$ values between empirical and regionalized FDCs have an average of 0.81 with a standard deviation of 0.26. The top-kriging weights could acceptably regionalize FDCs even in the case that overall agreement between empirical and interpolated TNDs is relatively low. However, if a target catchment is very isolated or has distinguishable physical differences from donor catchments, a caution is required because the proximity-based FDC transfer possibly yields biased flow quantiles. A further study is necessary to clarify predictive performance of the geostatistical FDC regionalization in relation to performance of the TND interpolation, though it is an outside scope of this study.

For calibration of GR4J model with the regionalized FDCs, we selected 28 catchments that showed acceptable $NSE_Q$ (>0.6) in the GR4J calibration and $NSE_{FDC}$ (>0.8) in FDC regionalization. The $NSE_{FDC}$ values of the 28 catchments had an average of 0.93. The overall comparison between observed and regionalized quantile flows is illustrated as a 1:1 scatter plot in Fig. 5b. The agreement between observed and estimated flow quantiles in Fig. 5b confirms that the geostatistical FDC regionalization is a practical and convenient method to bring powerful performance for estimating FDCs of ungauged catchments where topological proximity adequately explains spatial patterns in flow quantiles.



### 4.3 Calibration of GR4J with regionalized FDCs

With the Monte Carlo simulations, we searched GR4J model parameters that reproduce the regionalized flow quantiles. $NSE_{FDC}$ values the between simulated and regional FDCs were greater than 0.90 with a range of 0.90-0.99 for the 28 selected catchments. Based on the high $NSE_{FDC}$ values, 10,000 simulations would be adequate to find parameter sets that acceptably

regenerate FDCs in lieu of a sophisticated optimization algorithm such as the SCE.

Fig. 6 compares observed hydrographs with those simulated with the parameter sets fitted to the regional FDCs for the same catchments in Fig. 4. We had an indication that the model calibration with a FDC could be competitive with the conventional calibration with observed flows (Fig. 6a) but of higher uncertainty. $NSE_Q$ of the catchment 15 (Fig. 6b) significantly decreased from 0.65 to 0.03 despite the relatively small change in $NSE_{FDC}$. $NSE_Q$ values of the 28 catchments ranged

between -0.173 and 0.852 with an average of 0.538 when calibrated with regional FDCs. This relatively wide range of $NSE_Q$ implies that parameter sets with high reproducibility of flow quantiles could be insufficient to predict both flow magnitude and timing although FDCs have worked well as a single criterion for model calibration in several case studies (e.g., Westerberg et al., 2011; 2014). It may be attributed to error sources in regionalized FDCs, but those were limited by selection of the catchments with high $NSE_{FDC}$ values. From this result, it is suggested that calibration of runoff model with a

FDC can be good for regenerating frequency of observed flows, but likely to have higher uncertainty particularly in flow timing than hydrograph-based calibration.

### 4.4 Evaluation of two proximity-based approaches for ungauged catchments

We compared the two approaches: (a) transferring model parameters from neighbouring gauged catchments (referred to as PROX_reg hereafter); (b) calibrating model parameters with regionalized FDCs (referred to as RFDC_cal hereafter). The 28

catchments selected for RFDC_cal were applied for leave-one-out cross-evaluations of the approaches.

While we made the evaluations with the chosen 28 catchments, 34 catchments with $NSE_Q > 0.6$ were used to transfer parameter sets to the 28 target catchments. 6 catchments were used for transferring parameters only, but not included in the evaluations. Likewise, despite all empirical FDCs of the 45 catchments were used for the geostatistical regionalization, 17 catchments with low $NSE_{FDC}$ (< 0.8) were unselected for the cross-evaluations. A hypothesis behind the cross-evaluations,

hence, is that neighbouring gauged catchments have acceptable predictive performance in runoff modelling and FDC regionalization.

### 4.4.1 Evaluation of hydrograph predictability

Performance of the two approaches for predicting flows is evaluated in terms of $NSE_Q$ and $VE_Q$, as summarized in Fig. 7. It is clearly recognized that PROX_reg outperforms RFDC_cal in prediction of flow magnitude and timing from the $NSE_Q$

comparison (Fig. 7a), whereas RFDC_cal was slightly better for estimating runoff volume (Fig. 7b). The both methods had higher medians of $NSE_Q$ values for the calibration period than the validation period, and this may indicate that parameters





fitted to flow observations or signatures for a certain period is better to be regionalized for the same period. However, catchments with increasing $NSE_Q$ values from calibration to validation periods were found as many as those with decreasing $NSE_Q$. It is thus difficult to conclude that model parameters calibrated to observed flows for a certain period would work better for the same period in ungauged catchments. Instead, more uncertainty would be involved in predicted flows if

calibration and regionalization periods do not match. From the $NSE_Q$ comparison, it is recognized that PROX_reg has higher performance and less uncertainty than RFDC_cal in flow prediction.

On the other hand, in $VE_Q$ comparison (Fig. 7b), RFDC_cal showed slightly smaller medians and shorter box heights than PROX_reg (Fig. 7b). $NSE_Q$ could be over-sensitive to peak flows, hence the entire flow regime might not be captured even with high $NSE_Q$ (Hrachowitz et al., 2013). The $VE_Q$ comparison indicates that reproducing regionalized FDCs could be

better for minimizing long-term bias in flow prediction than parameter regionalization. The median of $VE_Q$ increased from calibration to validation periods in the both cases. As was the $NSE_Q$ comparison, less predictive and more uncertain runoff volumes could be generated if parameters are temporally transferred. From the $NSE_Q$ and $VE_Q$ comparison, we had an indication that PROX_reg would be better for prediction in flow magnitude and timing, while RFDC_cal is good to reduce long-term bias in predicted flows.

**4.4.2 Evaluation of FDC predictability**

FDC reproducibility of the two methods was evaluated in terms of $NSE_{FDC}$, and is summarized in Fig. 8. Despite additional error sources introduced by input data and model structural deficiency (Hrachowitz et al., 2013), we realized that a parameter regionalization could have comparable performance for predicting FDCs to a direct regionalization of flow quantiles. In Fig. 8a, 13 catchments showed higher $NSE_{FDC}$ values with PROX_reg than those with the geostatistical FDC regionalization

albeit a less median of PROX_reg. It implies that the error sources in parameter regionalization could be less significant than those in direct regionalization of FDCs in some cases. The $NSE_{FDC}$ values of regionalized FDCs and RFDC_cal had similar medians, but the range of RFDC_cal was slightly wider (Fig. 8b) because parameter uncertainty and structural model deficiency are additionally introduced in RFDC_cal. From Fig. 8a and 8b, it is known that RFDC_cal could be somewhat better than PROX_reg for FDC prediction, but PROX_reg is also of strong predictive performance for FDCs.

As was in the $NSE_Q$ and $VE_Q$ comparison, temporal transfer of parameters from one to another period would add uncertainty in FDC prediction (Fig. 8c and 8d). In addition, RFDC_cal seems to provide more stationary prediction of FDCs than PROX_reg based on the higher median and narrower range of $NSE_{FDC}$ values.

Briefly, direct FDC regionalization or RFDC_cal would not guarantee higher performance in FDC prediction than parameter regionalization. PROX_reg in this study also had comparable performance in reproducing FDCs to direct regionalization of

FDC and RFDC_cal. Hence, the two-step approach of RFDC_cal, i.e., regionalization of empirical FDCs and model calibration, would be less pragmatic than simple PROX_reg that requires transferring parameters only.



### 4.4.3 Evaluation of flow signature reproducibility

Fig. 9 summarizes performance of PROX_reg and RFDC_cal for regenerating flow signatures in terms of $R_{QP}$, $I_{BF}$, and $R_{LD}$. RFDC_cal outperformed PROX_reg in $I_{BF}$ prediction whereas PROX_reg showed better predictability in $R_{LD}$. RFDC_cal was expected to perform for reproducing $I_{BF}$ based on its less bias in runoff volume because baseflow accounts for much

longer period within annual hydrographs. The high $R_{LD}$ reproducibility of PROX_reg is also expected from the high $NSE_Q$ values in flow simulations, which indicates good prediction in timing of peak flows.

In the case of $R_{QP}$, RFDC_cal appears to be slightly better than PROX_reg as does in the $VE_Q$ comparison. The bias in simulated flows mainly affects $R_{QP}$, and thus $R_{QP}$ comparison would be similar to the $VE_Q$ evaluation. This result is similar to the evaluation of FDC predictability, which indicated a slightly better performance of RFDC_cal.

## 5 Discussion

### 5.1 Predicting FDCs of ungauged catchments using parameter regionalization

A direct regionalization of empirical FDCs has been generally preferred to predict FDCs in ungauged catchments because of unavoidable error sources in parameter regionalization, i.e., model deficiency and uncertainty in input data and parameters. The strong performance of PROX_reg in this study, however, needs to be underlined. Fig. 8a suggests that FDCs of

generated hydrographs with transferred parameters can be comparable to direct regionalization of FDCs. The geostatistical FDC regionalization used in this study is a promising method with strong predictive performance (Pugliese et al., 2014; 2016), and poorly predicted FDCs were unselected in the cross-evaluations. Indeed, PROX_reg was only based on spatial proximity between the catchments for transferring GR4J parameters. Under these conditions, PROX_reg performed substantially ($\overline{NSE_{FDC}} > 0.90$). It implies that use of parameter regionalization may become the best option for predicting

FDCs if an ungauged catchment is surrounded by well-modelled gauged catchments. Because hydrographs are more informative than FDCs, PROX_reg may be preferable in both hydrograph and FDC predictions.

Since FDCs reflect collective influences of climate, vegetation, soil, and topography on streamflow, a small selection of physical properties or only topological proximity could inadequately explain spatial variation of observed flow quantiles. FDC regionalization could hence include substantial errors from unexpected spatial patterns such as land use changes.

Although our comparative analysis may be an exceptional case, it is unlikely to be general that parameter regionalization will have more errors in FDC prediction than a direct FDC regionalization (Hrachowize et al., 2013). However, there exist preconditions for applying parameter regionalization. Runoff models employed for gauged catchment should have a minimized concern of over-parameterization because it leads to model equifinality and large prediction errors. The regionalization method chosen for ungauged catchments also should be suitable for regional characteristics. The proximity-

based parameter regionalization has performed well under humid climates while regression- or similarity-based methods





could be better under arid conditions (Parajka et al., 2013). It is also notable that direct FDC regionalization slightly outperformed PROX_reg on average.

## 5.2 Use of regional FDCs as a single criterion for calibration

Empirical or regionalized FDCs have been adopted as a single measure or additional constraints for model calibration (e.g., Westerberg et al., 2014; 2011; Kapangaziwiri et al., 2012; Yadav et al., 2007; Yu and Yang, 2000), and led to enhanced flow prediction in ungauged catchments. Despite the successful applications, regionalized FDCs possibly cause high prediction errors in flow timing when employed as a single criterion for calibration. The uncertainty in parameter regionalization would be less significant than that from no flow timing in regionalized FDCs. Since we only chose regional FDCs with high $NSE_{FDC}$ values for the cross-evaluations, errors in the FDC regionalization were unlikely to be concerns.

The low $R_{LD}$ reproducibility of RFDC_cal in Fig. 9 also implies that use of regionalized FDCs as a single measure can yield substantial errors particularly in timing of peak flows. FDCs or metrics from FDCs such as slopes and TNDs of the FDCs could represent behavioural function of catchments, but could be less practical for predicting hydrographs than regionalized model parameters. A relative strength of RFDC_cal was low bias in runoff volume and baseflow predictions, and therefore RFDC_cal is recommended when modelled hydrographs are used for a low flow analysis or a long-term water resources management rather than a flood risk analysis.

## 5.3 Suggestions for improving flow prediction in ungauged catchments

The flow signature reproducibility depicted in Fig. 9 is insightful information for improving flow prediction in ungauged catchments. As evaluated, RFDC_cal has strength in runoff volume and baseflow prediction while flow magnitude and timing were well captured by PROX_reg. The gap in baseflow estimation of PROX_reg thus can be filled by incorporating FDCs into parameter calibration and regionalization (e.g., Pfannerstill et al., 2016). Unobservable conceptual parameters fitted to hydrographs may not be unique sets representing behaviour of gauged catchments, and eventually lead to the typical equifinality problem. If FDCs are used as orthogonal information conditioning the calibration process, the uniqueness of calibrated parameters will be enhanced and eventually lead to less uncertainty in the regionalization. If a regional calibration (e.g., Parajka et al., 2007; Kim and Kaluarachchi, 2008) included FDCs or $I_{BF}$ as additional constraints, the regional model would work better for capturing spatial patterns of parameters.

RFDC_cal, on the other hand, can evolve through conditioning the FDC-based calibration with additional flow metrics. The regional FDCs would provide less unique parameter sets than regionalized parameters, and the gap is likely to be in timing of peak flows. Westerberg et al. (2014) pointed out that a regionalized FDC is useful as a basic flow signature; nevertheless, further conditioning with other flow signatures is required for reducing uncertainty in the prediction. Given high performance in baseflow of RFDC_cal, additional flow signatures would be orthogonal if they enable to improve predictability in flows timing. From the comparative analysis of this study, $R_{LD}$ can be a good candidate to enhance peak flow prediction of the FDC-based calibration.



## 6 Summary and conclusions

Flow prediction in ungauged catchments has been comprehensively explored and continually improved by a plethora of case studies, but still has gaps that should be filled. In this study, we compared two methods for predicting daily streamflow in ungauged catchments through leave-one-out cross evaluations of 28 catchments in South Korea. We began with parameter calibration of GR4J model with streamflow observations in 45 gauged catchments. As a counterpart, we employed a proximity-based geostatistical method for FDC regionalization, and selected 28 catchments that showed high performance in the model calibration and the FDC regionalization. The two methods were evaluated in terms of hydrograph, FDC, and signature reproducibility. The key findings from the comparative analysis are summarized as follows:

(1) The consistency between model input and output data is an important prerequisite for runoff modelling. The CPI was in a clear positive relationship with the performance criterion, $NSE_Q$, when calibrating GR4J model for the 45 catchments. Data quality could significantly affect model performance in gauged catchments and thus parameter regionalization. A screening process excluding catchments with low consistency between the input and output would increase predictive performance of parameter regionalization.

(2) The geostatistical FDC regionalization showed acceptable performance in FDC prediction despite its low TND reproducibility. The top-kriging weights interpolating TNDs had high potential for estimating FDCs; however, it is notable that considering topological proximity only can bring biased estimates of flow quantiles. For improvement, it would be necessary to take into consideration physical differences between neighbouring catchments.

(3) The typical proximity-based parameter transfer was of strong performance to regenerate hydrographs and FDCs, and seems to outperform model calibration with regionalized FDCs. Although regionalized FDCs have potential for capturing functional behaviour of ungauged catchments, no information of flow timing in FDCs would cause great uncertainty in hydrograph simulations. The proximity-based parameter regionalization, on the other hand, had comparable performance in FDC prediction to the direct regionalization of empirical FDCs.

(4) Relative merits of model calibration with regional FDCs were low bias in simulated hydrographs and strong performance in baseflow prediction. RFDC_cal showed greater reproducibility in low flows than PROX_reg without logarithmic transformation of flow quantiles. The parameters fitted to regional FDCs are likely to incline the model to capture long-term flow regimes rather than peak flows.

(5) Adopting $R_{LD}$ as an additional constraint would improve model calibration with regionalized FDCs. A major weakness of RFDC_cal is low predictive performance in magnitude and timing of peak flows, and hence including the average time to peak in hydrographs would be an orthogonal condition to regionalized FDCs. Combination of $R_{LD}$ and regional FDCs for model calibration possibly improve signature-based hydrograph modelling in ungauged catchments.

In conclusion, we suggest that classical parameter regionalization is pragmatic for predicting hydrographs and flow frequency of ungauged catchments in South Korea because of its simplicity and powerful performance. Preconditions for acceptable performance typically are: (a) adequate predictive performance of the runoff model in gauged catchments, and (b)



suitability of the regionalization method employed. A FDC is a representative flow signature presenting behavioural functions of catchments, and well-regionalized FDCs can be useful for model calibration; however, additional orthogonal information seems necessary to be comparable to traditional parameter regionalization. We found that $R_{LD}$ is a good candidate for improving signature-based model calibration, and believe that further studies on regionalization of relevant

flow signatures will improve runoff modelling in ungauged catchments.

**Acknowledgement**

This study was supported by APEC Climate Center. The authors send special thanks to Ms. Yoe-min Jeong and Dr. Hyungil Eum for their PRISM climate data sets.

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



Table 1: List of the gauged catchments and corresponding physical properties

| ID | Name | Lat[1] (°) | Area (km²) | Elv[2] (m) | NSE_Q[3] > 0.6? | N_FDC[4] > 0.8? | ID | Name | Lat (°) | Area (km²) | Elv (m) | NSE_Q > 0.6? | N_FDC > 0.8? |
|---|---|---|---|---|---|---|---|---|---|---|---|---|---|
| **1** | **Goesan Dam** | **36.64** | **676.7** | **363** | **Y** | **Y** | 24 | Chunyang | 34.88 | 144.9 | 201 | N | Y |
| 2 | Namgang Dam | 35.41 | 2293.4 | 431 | Y | N | 25 | Osu | 35.54 | 359.5 | 255 | N | Y |
| 3 | Miryang Dam | 35.50 | 103.5 | 512 | N | Y | **26** | **Daecheon** | **35.88** | **816.3** | **198** | **Y** | **Y** |
| 4 | Boryeong Dam | 36.29 | 162.3 | 244 | N | N | **27** | **Jeonju** | **35.76** | **275.9** | **176** | **Y** | **Y** |
| 5 | Buan Dam | 35.65 | 57.1 | 177 | N | N | **28** | **Hari** | **35.95** | **527.6** | **197** | **Y** | **Y** |
| **6** | **Seomjingang Dam** | **35.63** | **763.4** | **357** | **Y** | **Y** | 29 | Bongdong | 35.99 | 345.0 | 245 | N | N |
| **7** | **Soyanggang Dam** | **38.03** | **2783.3** | **634** | **Y** | **Y** | **30** | **Hannaedari** | **36.75** | **283.9** | **126** | **Y** | **Y** |
| **8** | **Andong Dam** | **36.88** | **1628.7** | **543** | **Y** | **Y** | **31** | **Suchon** | **36.61** | **224.0** | **94** | **Y** | **Y** |
| **9** | **Yongdam Dam** | **35.81** | **930.4** | **510** | **Y** | **Y** | **32** | **Wolpo** | **36.84** | **1157.8** | **315** | **Y** | **Y** |
| 10 | Imha Dam | 36.49 | 1975.8 | 388 | N | N | **33** | **Jeomchon** | **36.66** | **615.3** | **371** | **Y** | **Y** |
| **11** | **Hoengseong Dam** | **37.58** | **207.9** | **436** | **Y** | **Y** | 34 | Sancheong | 35.51 | 1130.8 | 554 | Y | N |
| **12** | **Habcheon Dam** | **35.71** | **928.9** | **495** | **Y** | **Y** | 35 | Seonsan | 36.10 | 988.4 | 298 | Y | N |
| **13** | **Chungju Dam** | **37.28** | **6705.1** | **608** | **Y** | **Y** | 36 | Nonsan | 36.18 | 476.6 | 151 | Y | N |
| 14 | Juam Dam | 34.94 | 1029.4 | 269 | Y | N | 37 | Ugon | 36.28 | 133.9 | 39 | N | N |
| **15** | **Jangheung Dam** | **34.79** | **192.3** | **198** | **Y** | **Y** | 38 | Seokdong | 36.24 | 155.9 | 71 | N | Y |
| **16** | **Jungranggyo** | **37.70** | **208.6** | **131** | **Y** | **Y** | 39 | Cheongju | 36.58 | 165.3 | 149 | Y | N |
| **17** | **Munmak** | **37.44** | **1137.7** | **303** | **Y** | **Y** | **40** | **Heodeok** | **36.26** | **609.3** | **193** | **Y** | **Y** |
| **18** | **Yeongchun** | **37.38** | **4775.2** | **996** | **Y** | **Y** | **41** | **Yuseong** | **36.25** | **246.0** | **193** | **Y** | **Y** |
| **19** | **Yeongwol-1** | **37.42** | **1613.5** | **625** | **Y** | **Y** | 42 | Boksu | 36.19 | 161.6 | 216 | N | Y |
| **20** | **Pyeongchang** | **37.54** | **695.6** | **720** | **Y** | **Y** | **43** | **Sangyeogyo** | **36.44** | **495.0** | **255** | **Y** | **Y** |
| **21** | **Naerincheon** | **37.88** | **1013.0** | **752** | **Y** | **Y** | 44 | Gidaegyo | 36.47 | 361.2 | 250 | N | Y |
| **22** | **Wontong** | **38.17** | **299.7** | **707** | **Y** | **Y** | **45** | **Indong** | **36.27** | **68.0** | **203** | **Y** | **Y** |
| **23** | **Hampyeong** | **35.13** | **105.2** | **87** | **Y** | **Y** | | | | | | | |

[1]Latitude. [2]Mean Elevation. [3]NSE_Q in calibration with observed hydrographs. [4]NSE_FDC of regionalized FDCs. The catchments in bold were used for the leave-one out cross-evaluations.



**Table 2: Ranges of GR4J model parameters used for calibration**

| Parameter | 80% confidence interval (Perrin et al., 2003) | Range for calibration |
|---|---|---|
| X1 (mm) | 100-1200 | 10-2000 |
| X2 (mm) | -5-3 | -8-6 |
| X3 (mm) | 20-300 | 10-500 |
| X4 (days) | 1.1-2.9 | 0.5-4.0 |





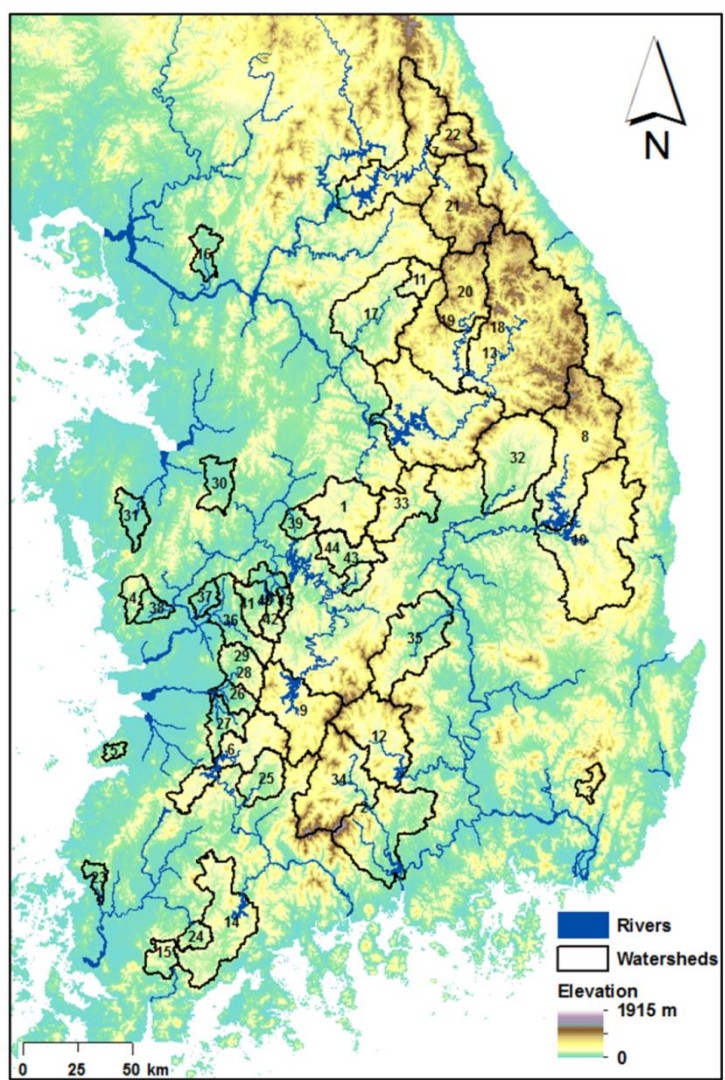

**Figure 1: Locations of the gauged catchments for GR4J model and FDC regionalization. Catchment numbers are labelled at the centroid of each catchment.**





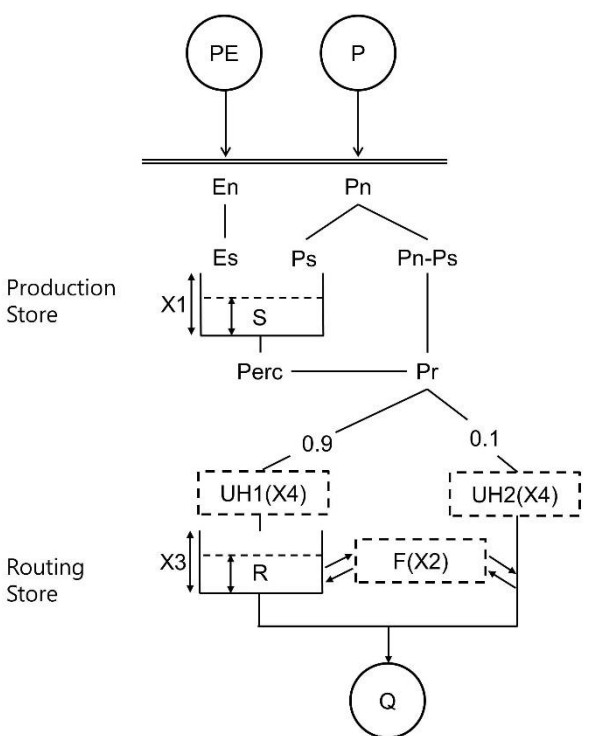

**Figure 2: The schematized structure of GR4J (X1-X4: model parameters, PE: potential evapotranspiration, P: precipitation, Q: runoff, other letters indicate variables conceptualizing internal catchment processes).**





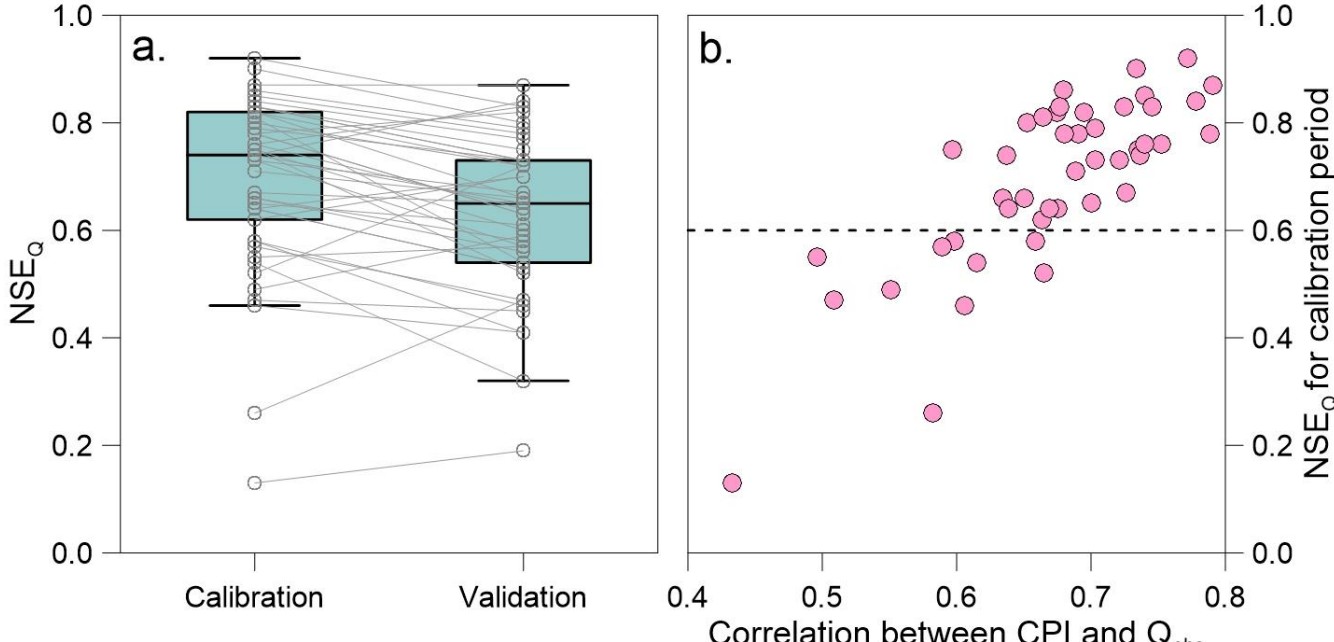

**Figure 3: Distributions of NSE$_Q$ values achieved from GR4J calibrations with observed hydrographs (a), NSE$_Q$ versus correlation coefficients between CPI and Q$_{obs}$ (b). Straight lines in (a) connect two NSE$_Q$ values for the calibration and validation periods of each catchment.**





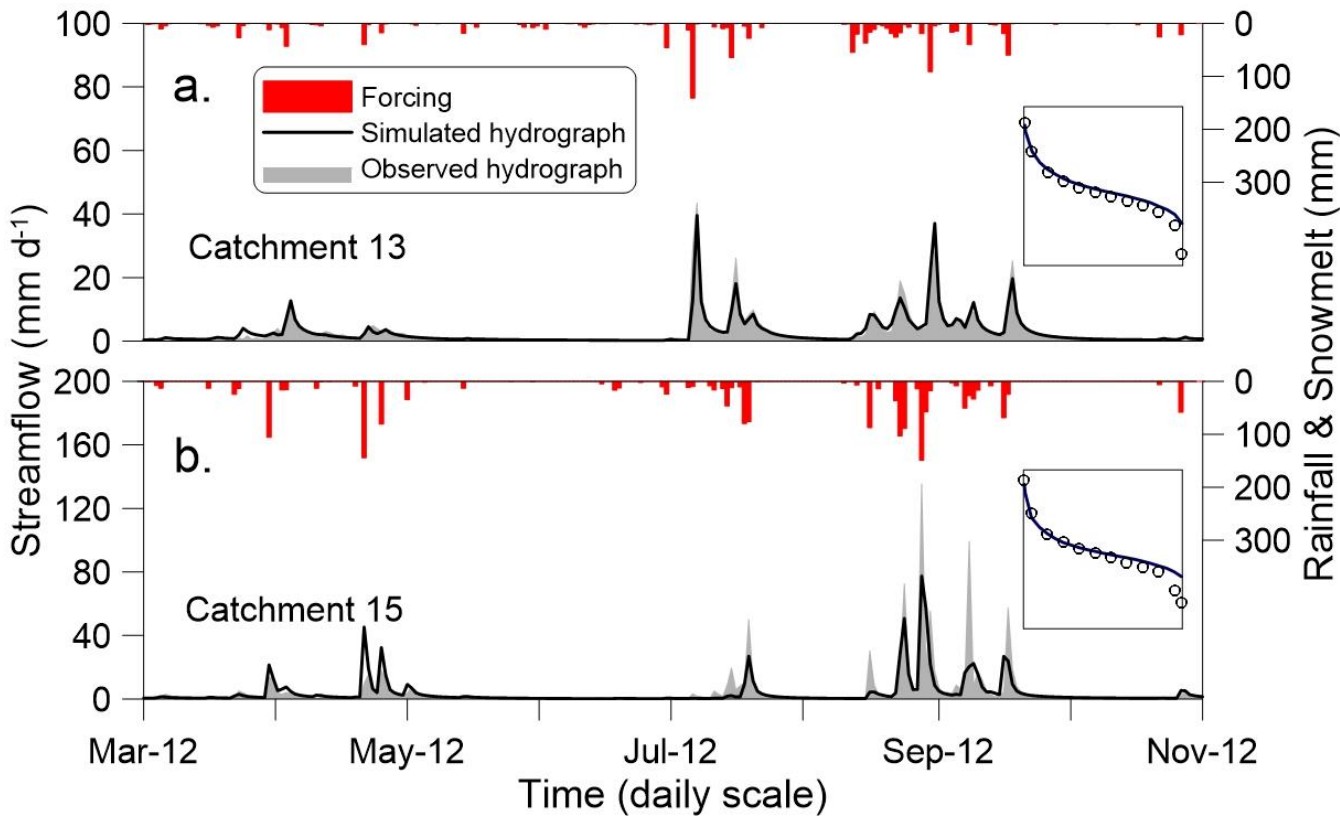

**Figure 4: Example hydrographs of two selected catchments that showed a high $NSE_Q$ (a) and a relatively low $NSE_Q$ (b) in model calibration using observed hydrographs. $NSE_Q$ values for the calibration period are 0.87 (a) and 0.64 (b) while $NSE_{FDC}$ values of the both cases are greater than 0.95. Line and scatter plots inside of the hydrographs depicts simulated (lines) and observed (circles) FDCs (quantile flows are in logarithmic scale).**





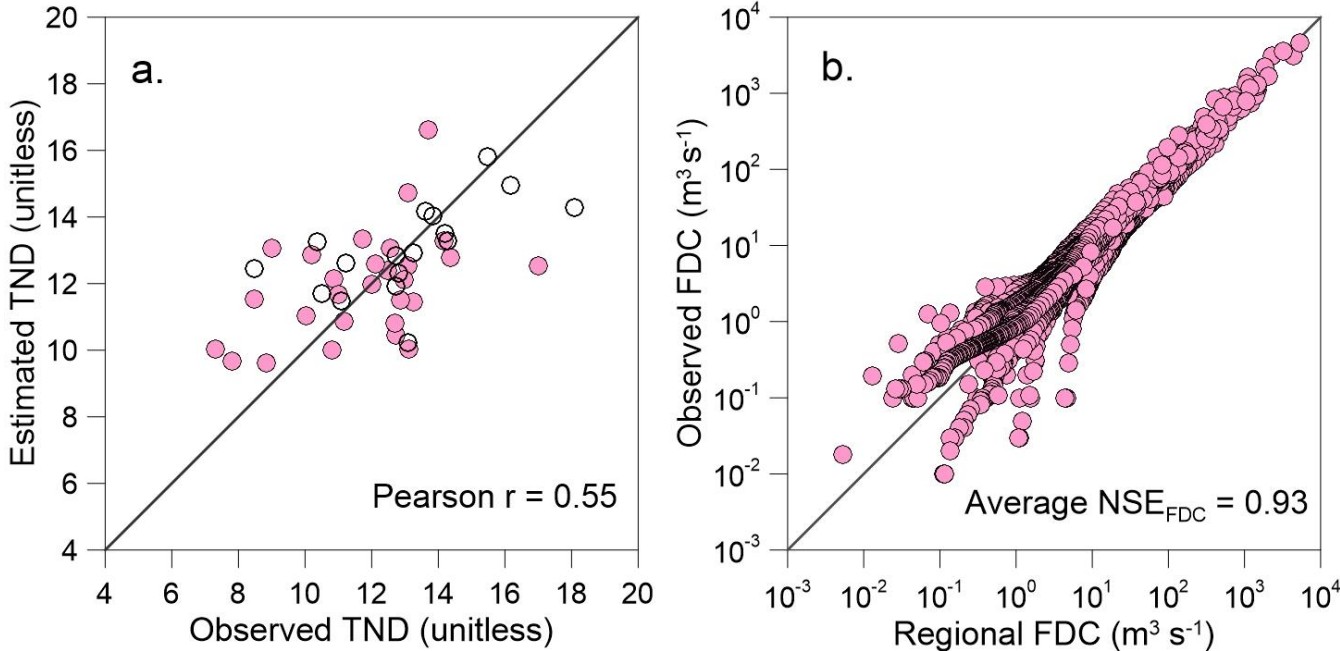

**Figure 5: 1:1 scatter plots of the observed and estimated TNDs (a), and the observed and estimated quantile flows of 28 catchments selected for RFDC_cal (b). The uncoloured symbols indicate unselected catchments for RFDC_cal.**





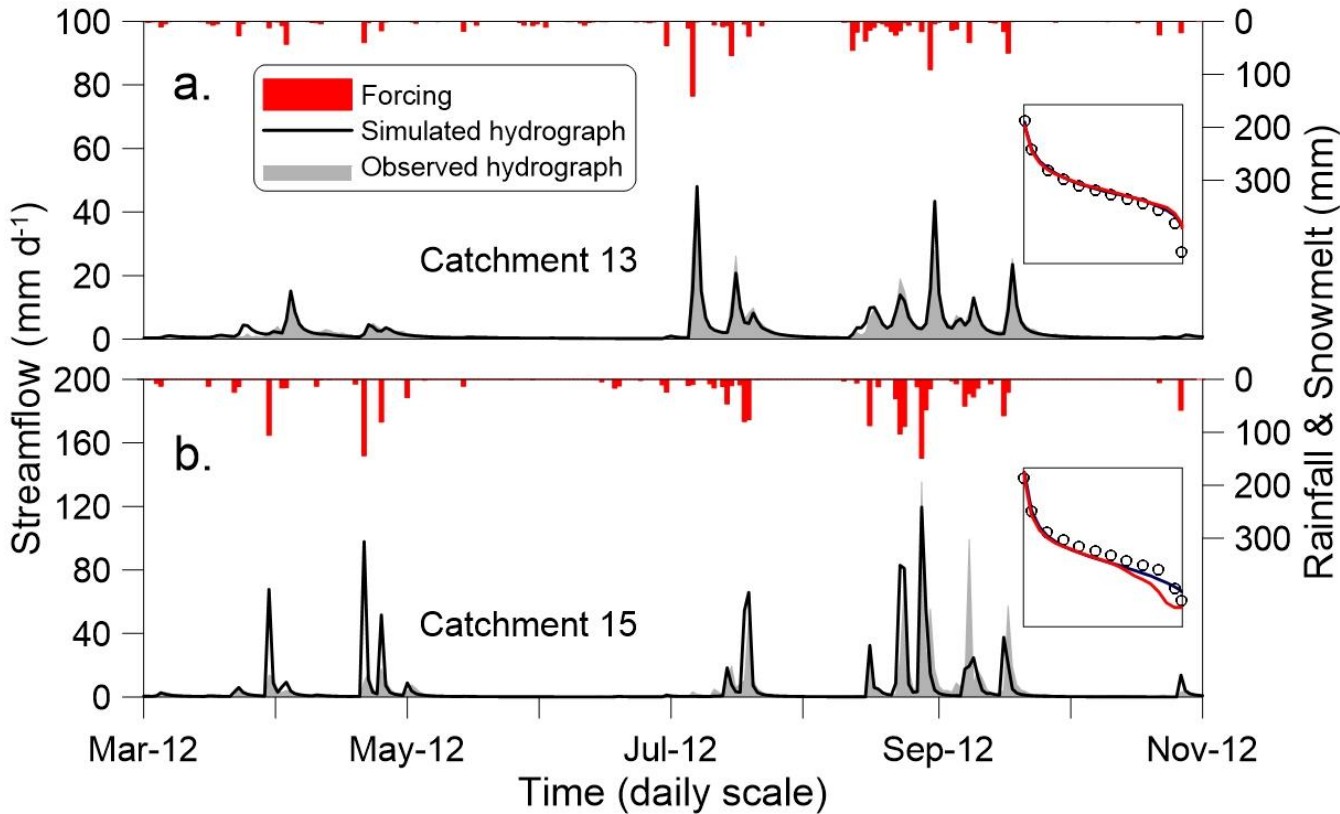

**Figure 6: Example hydrographs simulated by parameter sets calibrated with the regionalized FDCs. Line and scatter plots inside of the hydrographs depicts simulated (navy blue lines), regionalized (red lines) and observed (circles) FDCs (flow quantiles are in logarithmic scale). $NSE_Q$ values for the calibration period are 0.85 (a) and 0.03 (b), while $NSE_{FDC}$ between simulated and observed FDCs are 0.99 (a) and 0.76 (b). $NSE_{FDC}$ between simulated and regional FDCs are greater than 0.95 in the both cases.**





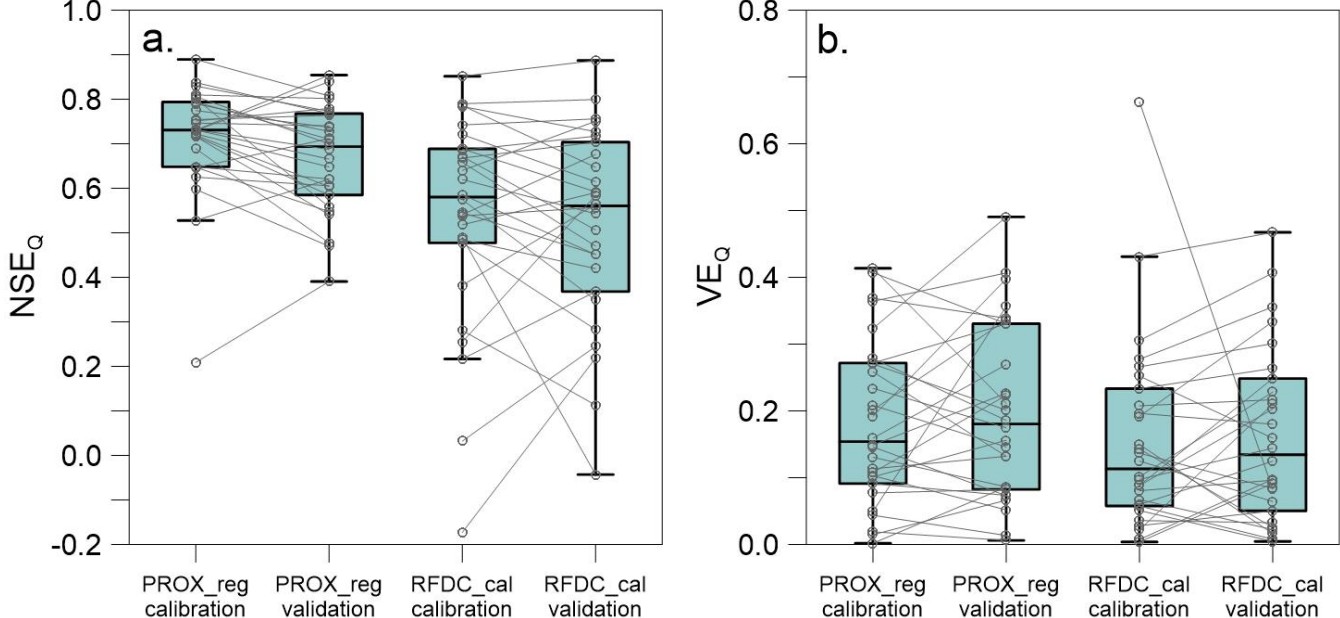

**Figure 7: Distributions of NSE$_Q$ (a) and VE$_Q$ (b) values for PROX_reg and RFDC_cal. The performance indicators for the calibration and validation periods of each catchment are linked with the straight lines.**





**Figure 8:** NSE$_{FDC}$ **comparison between PROX_reg and FDC regionalization (a), RFDC_cal and regionalized FDCs (b), the calibration and validation periods of PROX_reg (c), and the calibration and validation periods of RFDC_cal (d). RegFDC indicates values obtained from the geostatistical regionalization. The straight lines link the performance indicators for the**
5  **calibration and validation periods of each catchment.**





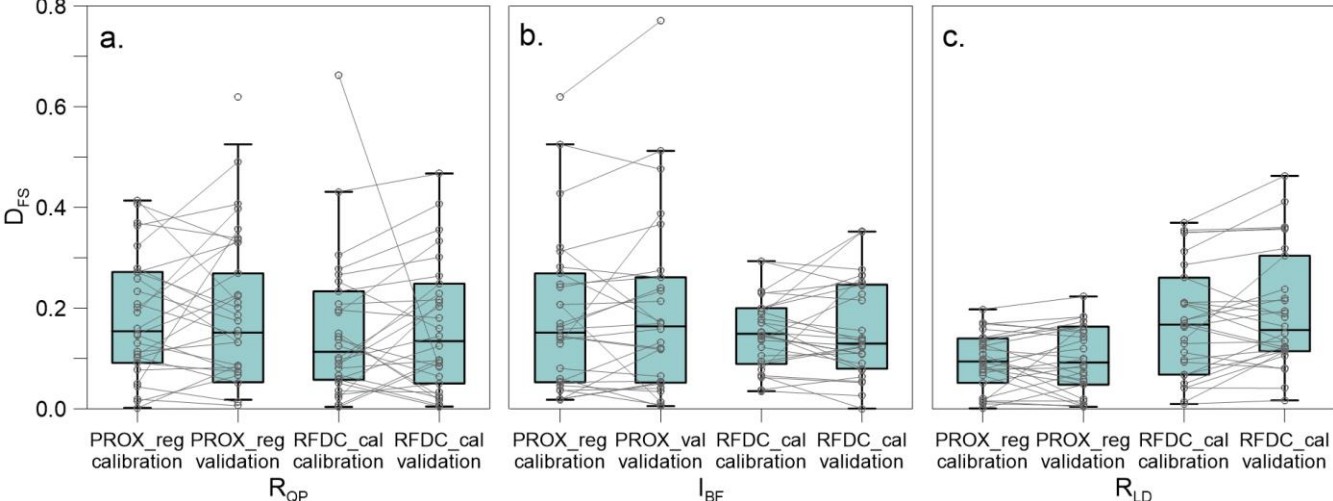

**Figure 9: Distributions of relative absolute error between observed and simulated three flow signatures, $R_{QP}$ (a), $I_{BF}$ (b), and $R_{LD}$ (c), for PROX_reg and RFDC_cal respectively. The straight lines link the performance indicators for the calibration and validation periods of each catchment.**