# Peer review of "A comparison between parameter regionalization and model calibration with flow duration curves for prediction in ungauged catchments"

_Hydrology and Earth System Sciences, 2016_

## Referee Comment (RC1) · Anonymous Referee #1 · 26 Nov 2016

This study compares two regionalisation methods for runoff predictions: the traditional spatial proximity and the model calibration against regionalised flow duration curves. I found this study has a very limited contribution to the predictions in ungauged basins (PUB). The major reasons are as follows:

1. There are numerous studies carried out by using various methods for PUB. The hydrological modelling uses spatial proximity, physical similarity and regression to regionalise its calibrated parameter sets to ungauged catchments. There are several studies that used flow duration curve methods for runoff predictions (Shu and Ouarda, 2012; Zhang et al., 2015), which just use observed runoff and catchment attributes to predict runoff time series, but does not need to involve any hydrological modelling.

This kind of research is totally ignored by the authors. These researches use the three steps to predict daily runoff: (1) building FDC method (geostatistical methods, statistical methods, etc); (2) estimating flow quantile based on some assumptions; (3) predicting runoff time series. The predictions results are very impressive (see Zhang et al., 2015). 2. It is not surprise at all to see the calibration against regionalised flow duration curves performs worse than the traditional spatial proximity approach for runoff time series predictions since it does not include any runoff timing information, which is the key for runoff time series predictions. 3. The sample number used here is too small. There are only 45 catchments used to evaluate regionalisation skill. Therefore, it is hard to get a generalised conclusion. Moreover, the authors only picked up 28 with good calibration of GR4J and FDC, making the sampling number extremely small. 4. The objective function. The selection of objective functions has very important implication on the conclusions. The authors used the classic NSE for hydrological modelling calibration/regionalisation. It will be inevitable that the predictions from spatial proximity regionalisation are better for high flow, but poorer for low flow. For a comprehensive evaluation, an objective function that compromises high flow and low flow (i.e. Box-Cox transformed streamflow) should be used.

Based on the above-mentioned reasons, I do not recommend this manuscript to be published in the prestigious hydrological journal: HESS.

Follows are specific comments: 1. Introduction. It is not comprehensive. Lots of methods used for building FDC in ungauged catchments are not introduced. Lots of studies using FDC to predict runoff time series are ignored (see the above-mentioned are just some examples). Lots of spatial proximity regionalisation studies are not included. The authors should have a comprehensive literature review from ISI Web of Knowledge. 2. It is very confused for the streamflow gauges used. The authors state that the 45 streamflow gauges used in this study are with negligible regulations (river diversion and dam operation), but they also state that " . . . operationally recorded at 16 multi-purpose dams for the Water Resources Management Information . . ." Is it really all the

gauges are with negligible regulation? 3. Are all gauges not nested? Please clarify. 4. Cross-validation and regionalisation. I am not against the cross validation (2011-2015 for model calibration and 2007-2010 for model cross validation). For regionalisation, I suggest to use the full period of dataset. Bury in mind, there are only nine years data for each gauge. 5. GR4J requires precipitation and potential evapotranspiration for model inputs. It is not clear how the potential evaporation is calculated. 6. Objective function. To have a comprehensive evaluation of these two methods, please also include a Box-Cox transformed streamflow objective function. 7. First paragraph in section 3.3.1. Please include more references for the three regionalisation approaches. 8. Equation (4). How is the constant 3.171 * 10ˆ-5 derived? 9. The methods mixed with results. Half of section 4.2 should be moved to methodology 10. Use all gauges for regionalisation. Please use all 45 gauges for the regionalisation. It makes no senses to me to exclude the 17 catchments with low NSEFDC (<0.80). You can setup prerequisite for PUB. It is not fair for another approach. 11. Figure captions. It is hard to follow figure captions. Please spell out all the abbreviations. I spent lots of time to figure out these abbreviations.

References: Shu, C., Ouarda, T., 2012. Improved methods for daily streamflow estimates at ungauged sites. Water Resour. Res. 48, W02523. http://dx.doi.org/10.1029/2011wr011501.

Zhang YQ, Vaze J, Chiew FHS, Li M, (2015). Comparing flow duration curve and rainfall–runoff modelling for predicting daily runoff in ungauged catchments. Journal of Hydrology, 525, 72-86.

.

---

## Short Comment (SC1) · 26 Nov 2016

The manuscript "A comparison between parameter regionalization and model calibration with flow duration curves for prediction in ungauged catchments" compares parameter regionalization techniques with FDC-based model calibration. My specific comments are listed below;

1. A number of studies have already been conducted regarding the comparison parametric and non-parametric methods for the regionalization of FDCs. Some of the studies are listed below;

Ganora D, Claps P, Laio F, Viglione A. 2009. An approach to estimate nonparametric flow duration curves in ungauged basins. Water Resour Res. 45. doi:10.1029/2008WR007472

Qamar MU, Azmat M, Cheema MJM, Shahid MA, Khushnood RA, Ahmad S (2016) Model swapping: a comparative performance signature for the prediction of flow duration curves in ungauged basins. J Hydrol 541:1030–1041. http://dx.doi.org/10.1016/j.jhydrol.2016.08.012

The authors are advised to go through these studies in order to familiarize themselves with the latest developments in the field of PUB. They further have to defend how their study is different from the already executed comparative studies? Frankly, I don't see much innovation, here.

2. The authors used dataset from 2007-2015 for model calibration and validation phases. Such dataset is not enough to hunch the flow trends. In such case, the modeling technique can be considered suitable for a particular time phase but cannot be generalized due to inadequate data length. Since the results generated by the proposed model for the entire study area are tested by using a LOOCV procedure. One solution to increase the data length is to consider one station as ungauged, removing it from the whole database and estimating FDC for that station with the proposed approach.

3. Line 15 reads, "Though combining a temperature index snowmelt model with GR4J can be an alternative approach, it increases the number of parameters and thus model uncertainty". How can increase of parameters make the technique uncertain? Increasing parameters increase complexity but it always betters efficiency. A thorough explanation is needed for this claim.

4. The periods and range of streamflow data (2007-2015) and climatic data (1973-2015) are not overlapping? Will it not be problematic? Moreover how was the range of climatic data used in calibration and validation phases of the model?

5. The equitation (4) seems to be generated by multivariate regression analysis. The authors never explained its generation, which is inevitable. How effective was this rescaling?

6. Equation 6 should be eliminated as it is already discussed above (equation 2). More over the performances indices need to increased. I suggest including mean absolute error and root mean square error.

7. Why only five nearest neighbors were used? Why not, say, 8 or 10 or 12? There has to be a reason for that. The authors are suggested to go through the following paper and study Figure 9 in detail in which Samaniego and Kumar (2010) selected nearest neighbors by observing the error generated by different number of neighbors.

Samaniego, L., A. Bárdossy, and R. Kumar (2010), Streamflow prediction in ungauged catchments using copula-based dissimilarity measures, Water Resour. Res., 46, W02506, doi:10.1029/2008WR007695.

8. The authors never discussed the complications involved in the implementation of each method. The discussion section should also compare the simplicity of each method in terms of implementation.

---

## Referee Comment (RC2) · Anonymous Referee #2 · 28 Nov 2016

The paper "A comparison between parameter regionalization and model calibration with flow duration curves for prediction in ungauged catchments" by Kim D., Jung I. and Chun, J. A. shows a comparative assessment of two methods for predicting daily streamflow series in ungauged catchments employing on the one hand the parameter regionalization of a rainfall runoff model, on the other hand the calibration of the same model using predicted flow duration curves. Even if the authors do not introduce any novel technique or method, the topic is of a wide interest in the hydrological field, thus I believe it might be suitable for publication in HESS after some minor improvements, which in my view the author might consider to take into account. Also, the paper is well written and is rather complete in all its sections.

[Figure]

**Minor comments**

1. The authors rely often on to NSE efficiencies to assess the reliability of each model, and they conclude that, given the good results obtained with $PROX_reg$, which is the regionalization of the parameters of the rainfall runoff model, that this model is preferable to the other given its capacity to reproduce the true idrograph in time. However, this conclusion seems to strongly contrast with other performance indices they deliver, e.g. VEQ, RQP, IBF, so I would soften a little the conclusions and would let the judgment be more flexible.

2. NSE is the most used performance index in literature and I agree with using it, but recently it has been criticized its capacity to understand how a model produce good result with low flows, while it emphasizes the capacity to understand high flows, so that NSE could end to be a biased index (the authors also recall this behavior at P11 L8). I wonder how the final performances vary by adopting LNSE, which is the same as NSE but taking the natural logs of streamflows. I assume that the lesser influence of the low flow regime prediction into NSE might somehow introduce a distortion of final judgment. In case the results are substantially different I would recommend the authors to make an effort in discussing these results as well.

3. I would suggest to change the title including the word "streamflow" somewhere. In this way, the topic is clearer. Perhaps "A comparison between parameter regionalization and model calibration with flow duration curves for prediction of streamflows in ungauged catchments" ?

4. Table 1 reports for each model two columns in which the author say whether or not NSE efficiencies are greater than 0.6 or 0.8 respectively. I think that would be of more interest for the reader to see all the values for each catchment as well as the cut-off at 0.6 or 0.8. For instance, they can report efficiencies for each

catchments and let the numbers above the cutoff in bold face. Reporting "Y" or "N" only might result uninformative and, at the very end, useless.

5. I would recommend to extend paragraphs 3.3.1 and 3.3.2 to introduce some more details of the two proposed approaches. Furthermore, I would move those two paragraphs before the evaluation indices adopted.

**Technical notes and misspellings**

1. P3 L7. "Siberian high pressure", perhaps is "low pressure".

2. P8 L19. Please add the word "between" in between the words "coefficients" and "CPI".

3. P10 L3. Please remove the article "the" between the words "values" and "between".

4. P10 L4. "Based on the high [. . .]" is perhaps "Based on the highest [. . .]"?.

5. I have not completely understood what the author mean with "orthogonal" referring to streamflow signatures, please consider to add some more details arguing what does this word mean for them into the context of the sentence.

---

## Referee Comment (RC3) · Anonymous Referee #3 · 8 Dec 2016

Review of the manuscript "A comparison between parameter regionalization and model calibration with flow duration curves for prediction in ungauged catchments" by Kim et al.

In this manuscript, Kim et al. compared two regionalization approaches to predict hydrographs and flow-duration curves (FDCs) in ungauged catchments in South Korea. The proximity based parameter regionalization outperformed the model calibration with regionalized FDCs (from top-kriging) for the simulation of hydrographs, whereas their performance was comparable regarding the prediction of FDCs. Given the relative simplicity and the good performance of the parameter regionalization, they recommended to use this approach for the prediction of runoff in ungauged catchments in South Korea.

I like the idea of the manuscript and read the manuscript with interest. However, I think the current level of the manuscript does not fulfill the criteria for publication in HESS. The manuscript partly introduces a new concept for the prediction in ungauged catchments. It also uses existing methods that should be discussed more critically. Relevant literature is not sufficiently introduced and discussed and references for methods or statements are sometimes missing. The structure of the paper could be improved by better separating methods, results and discussion – as it is now elements of these parts are mixed. To make the manuscript better readable I would also recommend to improve the English which suffers from grammatical errors.

I hope that the comments below will be helpful for the authors to improve their manuscript.

**Major comments:**

1. In this manuscript two regionalization approaches are compared: the first approach regionalizes parameter sets calibrated with the hydrograph, and the second approach regionalizes normalized FDCs that are used for the calibration of a runoff model. Although I like this second approach it seems to be rather different from the first one. I wonder why the authors did not apply an approach that is closer to the first one such as calibrating the model using the FDC and regionalizing these calibrated parameter sets. Using the suggested approach would make the results more comparable because the uncertainty sources are more similar (e.g. uncertainty due to top-kriging would be eliminated).

2. The manuscript would benefit from a detailed discussion on the sources and the influence of uncertainties related to the different regionalization approaches.

They are crucial for the interpretation of the results.

3. Besides the hydrograph and the FDC also runoff ratio, baseflow index and rising limb density of the ungauged catchments are evaluated. The authors state several times that the calibration of the runoff model against the regionalized FDC and the rising limb density simultaneously would improve the prediction in ungauged catchments. However, I see no strong evidence for this statement based on Fig. 9. I also don't understand why the rising limb density is regarded as being orthogonal to the FDC. I recommend to weaken these statements or to provide good evidence for it. Furthermore, no information is provided on how the rising limb density could be derived for the ungauged catchment. Would you also regionalize it?

4. It could be interesting if you actually tried to constrain the runoff model by the FDC and the rising limb density (or any other suitable runoff signature).

-kriging is used for the regionalization of the normalized FDCs. Is this approach really a well-established method as you mention? How many studies have used this approach? Is this approach suitable for FDCs and the density of the gauging stations in your study area? Can you give good reasons for not using ordinary kriging?

5. The results from the two regionalization approaches are presented as separate numbers (performance value) or separate boxplots that are next to each other (Fig. 7-9) which is inconvenient for their direct comparison. I would recommend to improve the presentation of the results by using the parameter regionalization approach as a benchmark to which the second approach (calibration with regionalized FDC) is compared. E.g. take the difference between the Nash-Sutcliffe efficiency of approach one and two for each catchment.

6. The snowmelt model used for the calculation of snow accumulation and ablation needs more explanation, especially because snowmelt models based on energy

balance usually are data intensive. From shortly reading the publication from Walter et al. (2005), I don't have the impression that this physics based snowmelt model is simple, as you call it. I agree that the snowmelt model doesn't have parameters that are calibrated, however it has various parameters that have to be estimated (e.g. cloud cover, albedo, windspeed, etc.). It would be worth to discuss whether there is really less uncertainty involved than when using e.g. a degree-day method for the simulation of snow accumulation and ablation.

**Moderate comments:**

P1 L16-18: I would remove the sentence about the rising limb density and instead add some information about the results of the FDC prediction in the ungauged catchments, because that was one focus of your study.

P2 L1: The study from Seibert and Beven (2009) did not use any regionalization in their analysis. This paper is not the right citation here. Please make sure that you cite properly.

P2 L7: When writing about regionalized flow signatures it would be worth to include the study of Yadav et al. (2007) and Hingray et al. (2010) at this point.

- Yadav, M., Wagener, T., and Gupta, H. V.: Regionalization of constraints on expected watershed response behavior for improved predictions in ungauged basins, Adv. Water Resour., 30, 1756– 1774, doi:10.1016/j.advwatres.2007.01.005, 2007.

- Hingray, B., Schaefli, B., Mezghani, A., and Hamdi, Y.: Signature-based model calibration for hydrological prediction in mesoscale Alpine catchments, Hydrolog. Sci.

P2 L9: You mention that flow signatures have been frequently applied for model calibration. Please give some more examples including runoff ratio, baseflow index and rising limb density.

P2 L16: Please give more examples of studies that regionalize FDCs and explain how they do it. This is important because the regionalization of FDCs is a core method of your study.

P2 L19-26: The information in the first sentence contradicts your subsequent paragraph.

P3 L4-11: Please cite where your information about this paragraph comes from. Is the information of this paragraph for South Korea in general or does it only relate to the study catchments?

P3 L13-L17: Where is the data of the 29 catchments with high quality from? Please add this reference and also the reference for the inflow data of the multi-purpose dams to the reference list at the end of the manuscript.

P3: Chapter on description of study area and data: where is the evaporation data from that you need as input for the runoff model? Do you need elevation data for the runoff model? Do you have any information about geology, vegetation etc. because you mention this as possible reason for poor top-kriging performance (P9 L15).

P4 L3-8: I recommend to add a short description of the structure of GR4J, information about the required input data and its resolution as well as information about the use of elevation bands.

P6 L24: Why do you evaluate the hydrograph with Nash-Sutcliffe and volume error, but the FDC only with Nash-Sutcliffe?

P6 L28: Give an explanation why you selected runoff ratio, baseflow index and rising limb density as signatures. Why three signatures?

P7 L22: Please explain why you use 5 donor catchments and not 3 or 7.

P8 L9: Again, why do you use 5 parameter sets and not 3 or 7? Does it make sense to give weights to these 5 parameter sets given the uncertainty related to them?

P8 L31: Why do you use NSE(Q) 0.6 as threshold?

P9 L4-10: This paragraph belongs to the methods section and should not be in the results. Do I understand correctly that 5 donors were used for the regionalization of the FDC? If yes, how do you get a total weight of 1? If no, please write this sentence more clearly.

P9 L29: Why a threshold of NSE(FDC) of 0.8 is used? Is it necessary to reduce the number of catchments used in model calibration by the additional constraint of NSE(FDC)? Wouldn't it be better to keep as many catchments as possible?

P9 L31: Figure 5b – how do you explain the scatter in the low flow?

P12 L10: Discussion: The manuscript would strongly benefit from a deeper and more extensive discussion of the results with other studies. Many statements appear in the discussion for which it is not clear where they are taken from - so please cite other studies properly (e.g. first sentence in discussion). There is no chapter in the discussion about the prediction of the hydrograph in ungauged catchments.

P14 L1: Summary and conclusions: I recommend to shorten this chapter. Point 1 (L9-13) is in my opinion no key finding of the study, point 2 (L14-17) is more an assumption than a result and point 5 (L27-30) is also rather a hypothesis than a result and should be formulated as possible further steps.

P15: Please add the information on data availability and author contribution.

P20 L1: If you want to show the parameter ranges used by Perrin et al. (2003), you should also argue why you use different ones. Since there are only 4 parameters the information could also be added in the text.

P24: Figure 4 – please add labels to the FDC-plot. Why did you select catchment 15 which is within the 50

P26: Figure 6 – The authors often mention that the calibration with the regionalized FDC results in hydrographs with poor timing. Such timing issues are not obvious in the plots of Fig. 6. Thus, I would recommend to show time periods or catchments where timing really is a challenge.

P27-29: Figures 7, 8 and 9: These plots all look very similar to me and I recommend to condense or reduce the information of these plots. In my opinion it is not necessary to show the calibration values, I would rather focus on validation efficiency because that's the tougher criteria. I also think it's not ideal to compare the regionalization approaches in this way and I recommend to use the concept of benchmarks for comparison: e.g. make the difference between the benchmark strategy (RFDC_cal) and the PROX_reg strategy. The use of benchmarks results in one single value which can easily be interpreted: e.g. positive values mean that PROX_reg is better than RFDC_cal.

P29: Figure 9: I like the idea of evaluating further signatures and using them as additional constraints in model calibration. However, this plot does not give enough evidence for the conclusions drawn. To show that RLD and RFDC_cal are uncorrelated different methods are needed. I am also not convinced that the additional use of RLD would improve model calibration with the regionalized FDC more than RQP.

**Minor comments:**

Please use the HESS guidelines for all abbreviations, so that all are done in a similar style as e.g. the abbreviation of the baseflow index. Please also write Figure 9 and not Fig. 9 when you refer to it at the beginning of a sentence.

P1 L1: I would adapt the title: ". . .and model calibration with regionalized flow-duration

curves. . ."

P1 L12: Shouldn't it say "Leave-one-out cross validation"?

P3 L1: Why do you consider the selected signatures as "major signatures"?

P3 L11: Please provide numbers for the percentage of precipitation falling as snow.

P5 L25: Where does this equation for calculating MAP* come from? Why do you need the constant?

P7 L14: I recommend to integrate this whole chapter in chapter 3.1, because it is about regionalization and not about evaluation.

P7 L19: This information is already in the introduction and is not needed in the methods part. Furthermore, you cite different studies here than in the introduction.

P7 L25: I recommend to integrate this whole chapter in chapter 3.2, because it is about regionalization and not about evaluation.

P8 L2: I don't think that the regionalized FDC is used as objective function. It is rather used as reference value against which model simulations are evaluated.

P8 L25: Can you say what the CPI was for these catchments that were poorly modelled?

P10 L8: What about the efficiencies of catchment 13?

P10 L19: Please introduce the abbreviations such as PROX_reg earlier in the manuscript, because e.g. Fig. 5b already uses abbreviations.

P19 L1: Table caption should be adapted because NSE(Q) and NSE(FDC) are not catchment properties.

P21: Figure caption - it's the catchments that are labeled in the center and not the numbers. Also skip ". . .for GR4J model and FDC regionalization"

P23: Figure 3a – I agree that it is important to know that the model is able to simulate runoff in most catchments. However I don't think that a boxplot is needed for that. The median and the range of the model performance in calibration and validation could also just be mentioned in the text.

---

## Author Comment (AC1) · 23 Dec 2016

Dear anonymous referee:

We thank for your comprehensive review on our manuscript and greatly appreciate your valuable time and contribution. We generally agree to your comments and recommendations, and want to improve our manuscript for publication in HESS. We agreed that our manuscript needs to be restructured for clearly showing scientific contribution. In our opinion, it would be better to address comparison between hydrograph-based and FDC-based calibrations at gauged catchments first, and then move to the ungauged case (i.e. parameter regionalization and calibration with regional FDCs). To evidently suggest orthogonal flow signatures, it would be better to provide actual results of FDC-

based calibrations in combination with the three signatures (i.e. runoff ration, baseflow index, rising limb density) at gauged catchments. In revision, we will provide in-depth discussion on performance and uncertainty in simulated flows.

However, we still want to focus on our main research objective to compare parameter regionalization and calibration with regionalized FDCs for ungauged catchments. Although you proposed regionalization of parameters fitted to empirical FDCs, the parameter sets calibrated to FDCs (i.e. only flow magnitudes) are likely to have more uncertainty than those fitted to hydrographs (i.e. flow timing and magnitudes). Hence, we expected that regionalization of parameters from empirical FDCs would be more uncertain than conventional regionalization approaches too. On the other hand, it is difficult to answer whether calibration with regionalized FDCs has less performance than parameter regionalization or vice versa. Through this comparison, we want to provide novel information for selection of methods for predictions in ungauged catchments.

From our knowledge, daily runoff prediction in ungauged catchments has been barely studied in South Korea in comparative ways. Thus, our manuscript will be also beneficial to expand spatial coverage of previous regionalization studies. Considering your comments and recommendations, we want to restructure our manuscript as follows:

(1) We will include more literatures about prediction in ungauged catchments in the introduction as you requested. Additional literature will be about FDC regionalization and signature-based calibration methods. We will also introduce the decade-long project of the IAHS in Prediction in Ungauged Basins (Blöschl et al., 2013; Hrachowitz et al., 2013) for providing comprehensive knowledge about rainfall-runoff modeling in ungauged catchments.

(2) We will restructure the manuscript from gauged to ungauged catchments. First, we will show results and discussion about predictive performance and uncertainty at gauged catchment of both hydrograph- and FDC-based calibrations. Then, we will

move to compare and discuss the parameter regionalization and calibration with regional FDCs.

(3) The part of signature reproducibility will be replaced with actual FDC-based calibration in combination with three flow signatures at gauged catchments. We still expect the rising limb density (RLD) would be orthogonal because it is information of flow timing that FDCs do not have. Actual results will be evidence of our conclusions.

Once again, we thank for your contribution, and please find our response as per your comment below.

1. In this manuscript two regionalization approaches are compared: the first approach regionalizes parameter sets calibrated with the hydrograph, and the second approach regionalizes normalized FDCs that are used for the calibration of a runoff model. Although I like this second approach it seems to be rather different from the first one. I wonder why the authors did not apply an approach that is closer to the first one such as calibrating the model using the FDC and regionalizing these calibrated parameter sets. Using the suggested approach would make the results more comparable because the uncertainty sources are more similar (e.g. uncertainty due to top-kriging would be eliminated).

-> Although two approaches appear to be similar, uncertainty sources involved in them are very different. The proximity-based regionalization is to transfer parameters fitted to more informative data (i.e. both flow timing and amount), but there is no calibration process for ungauged catchments. On the other hand, fitting parameters to regionalized FDCs has a calibration process for ungauged catchments but with less informative data (i.e. only statistical flow amounts). The main research question of this manuscript is: which one is better between (1) no calibration for target ungauged catchments but more informative calibration in gauged catchments and (2) direct parameter calibration for ungauged catchments but with less informative data? It is difficult to answer if regionalized parameter sets have greater uncertainty than calibrated parameters with

less informative data or vice versa. When embarking on this study, we believed that this question is more meaningful than regionalization of parameter sets from empirical FDCs. In our sense, it was likely that regionalizing parameters fitted to both flow magnitude and timing would be more reliable and thus of a better predictive skill than regionalizing ones fitted to flow magnitudes alone. For clearly showing the objective of this study, we would include a figure that schematizes two approaches. And, we will explain our research objectives more clearly in the introduction.

2. The manuscript would benefit from a detailed discussion on the sources and the influence of uncertainties related to the different regionalization approaches. They are crucial for the interpretation of the results.

-> We will do it in revision. It would better to comparatively discuss uncertainty associated in calibration against runoff and empirical FDCs. We will include a quantitative comparison between two approaches at gauged catchments. Then, we will move to ungauged catchments for providing in-depth discussion about uncertainty in parameter regionalization and calibration against regionalized FDCs.

3. Besides the hydrograph and the FDC also runoff ratio, baseflow index and rising limb density of the ungauged catchments are evaluated. The authors state several times that the calibration of the runoff model against the regionalized FDC and the rising limb density simultaneously would improve the prediction in ungauged catchments. However, I see no strong evidence for this statement based on Fig. 9. I also don't understand why the rising limb density is regarded as being orthogonal to the FDC. I recommend to weaken these statements or to provide good evidence for it. Furthermore, no information is provided on how the rising limb density could be derived for the ungauged catchment. Would you also regionalize it?

-> We wanted to introduce flow signatures that can potentially enhance the FDC-based calibration. In Fig. 9c, PROX_reg has much less medians and heights in box plots than RFDC_cal. It confirms that RFDC_cal has a shortcoming to reproduce the average

time to peak in runoff time series. Because the rising limb density (RLD) itself is a flow signature indicating flow timing, we suggested a calibration against FDC (flow magnitude) plus RLD (flow timing) would enhance the signature-based calibration. However, we agree that this inference is not evident. In revision, we will test this hypothesis at gauged catchments and will provide the results. We expect better predictive performance from addition of RLD in calibration. However, regionalization of RLD is another important research topic. We believe it would be better to address this in a separated paper. The top-kriging method can also be a candidate method for regionalization of RLD. However, there is no guarantee that the geographical interpolation will show similar performance to the FDC regionalization. RLD could be more sensitive to physical properties of catchments than FDC regionalization. Since the calibration against empirical FDC plus RLD at gauged catchments can be a potential evidence for ungauged catchment, we want to state regionalization of RLD as a further topic.

4. It could be interesting if you actually tried to constrain the runoff model by the FDC and the rising limb density (or any other suitable runoff signature). -kriging is used for the regionalization of the normalized FDCs. Is this approach really a well-established method as you mention? How many studies have used this approach? Is this approach suitable for FDCs and the density of the gauging stations in your study area? Can you give good reasons for not using ordinary kriging?

-> As replied to comment 3, we want to provide actual calibration results against empirical FDC plus other signatures in revision. The geographical method is recently proposed by Pugliese et al. (2014), thus it has not been frequently adopted in previous studies. However, its performance in the original study was 0.914 and 0.922 in terms of NSE and Log NSE between observed and predicted quantile flows in 18 Italian catchments, and the geostatistical method outperformed other two conventional regionalization methods. In Pugliese et al. (2016), this method was also compared with a regression-based method for 182 catchments in southeastern U.S., and found good predictive performance in both low and high flow estimates. Top-kriging can consider

topological features of watersheds while ordinary kriging cannot. In top-kriging, nested catchments have more weights for interpolation than adjacent catchments. For spatial interpolation of functional behaviors of catchments, top-kriging seems to be better than the ordinary kriging. Based on this information, we adopted this method that has a great merit to non-parametrically preserve features in FDC continuum.

5. The results from the two regionalization approaches are presented as separate numbers (performance value) or separate boxplots that are next to each other (Fig. 7-9) which is inconvenient for their direct comparison. I would recommend to improve the presentation of the results by using the parameter regionalization approach as a benchmark to which the second approach (calibration with regionalized FDC) is compared. E.g. take the difference between the Nash-Sutcliffe efficiency of approach one and two for each catchment.

-> We will provide better presentations for clear understanding. In revision, we will use a combined objective function between NSE and log VE for calibration as in Zhang et al. (2015) such that we can have a balance between high and low flows in parameter calibration. We will also apply this to the FDC-based calibration for consistency. Then, we will take differences for each catchment as advised. We agree that it will improve readability.

6. The snowmelt model used for the calculation of snow accumulation and ablation needs more explanation, especially because snowmelt models based on energy balance usually are data intensive. From shortly reading the publication from Walter et al. (2005), I don't have the impression that this physics based snowmelt model is simple, as you call it. I agree that the snowmelt model doesn't have parameters that are calibrated, however it has various parameters that have to be estimated (e.g. cloud cover, albedo, windspeed, etc.). It would be worth to discuss whether there is really less uncertainty involved than when using e.g. a degree-day method for the simulation of snow accumulation and ablation.

-> Although the snowmelt model of Walter et al. (2005) is not very simple, it requires temperatures and precipitation only. In our sense, it is difficult to say that the snowmelt model is data-intensive as is a typical physics-based snowmelt model. As you commented, physical parameters (e.g., albedo, transmissivity, cloud cover, etc.) are necessary for physical snowmelt modeling. Hence, Walter et al. (2005) mainly addressed how to estimate them only with precipitation and maximum and minimum temperatures. We will more clearly address uncertainty sources of the snowmelt model in revision. However, snowmelt is of minor influences on streamflow regimes in South Korea because summer season rainfall is dominant liquid forcing to catchments. Our intention was to reduce bias from no snow component in GR4J snowmelt, not to better simulate the snowmelt process. For justification, we will provide ratios of highest SWEs to annual precipitation for each catchment. The reason why we did not combine a temperature index (i.e. degree-day) with GR4J is to avoid interaction between the temperature index and GR4J parameters. It can worsen the equi-finality problem. We still want to maintain the parsimonious structure of GR4J. We agree that it is necessary to answer if the addition of the degree-day actually causes higher uncertainty than the physical snowmelt model. However, we believe that it is not meaningful in the case that most information for parameter calibration is in summer season hydrographs. It would be better to regard the physical snowmelt modeling as one choice for considering snow component with no additional parameters.

Moderate comments: P1 L16-18: I would remove the sentence about the rising limb density and instead add some information about the results of the FDC prediction in the ungauged catchments, because that was one focus of your study.

-> We agreed. We will improve the abstract.

P2 L1: The study from Seibert and Beven (2009) did not use any regionalization in their analysis. This paper is not the right citation here. Please make sure that you cite properly.

-> It would be a mistake. We will check if all references are properly cited again.

P2 L7: When writing about regionalized flow signatures it would be worth to include the study of Yadav et al. (2007) and Hingray et al. (2010) at this point. • Yadav, M., Wagener, T., and Gupta, H. V.: Regionalization of constraints on expected watershed response behavior for improved predictions in ungauged basins, Adv. Water Resour., 30, 1756– 1774, doi:10.1016/j.advwatres.2007.01.005, 2007. • Hingray, B., Schaefli, B., Mezghani, A., and Hamdi, Y.: Signature-based model calibration for hydrological prediction in mesoscale Alpine catchments, Hydrolog. Sci.

-> We agreed. They are informative references. We will cite them.

P2 L9: You mention that flow signatures have been frequently applied for model calibration. Please give some more examples including runoff ratio, baseflow index and rising limb density.

-> We will add more studies on signature-based model calibrations in the introduction.

P2 L16: Please give more examples of studies that regionalize FDCs and explain how they do it. This is important because the regionalization of FDCs is a core method of your study.

-> Yes. We will more comprehensively introduce studies on regional FDCs (e.g., Shu and Ouarda, 2012)

P2 L19-26: The information in the first sentence contradicts your subsequent paragraph.

-> We will globally recheck and improve all sentences.

P3 L4-11: Please cite where your information about this paragraph comes from. Is the information of this paragraph for South Korea in general or does it only relate to the study catchments?

-> It is general description of climatic and geophysical characteristics of South Korea.

[Figure]

South Korea is not a large country, thus can be common features of the study catchments. We will provide references from the Korean Meteorological Administration for this paragraph.

P3 L13-L17: Where is the data of the 29 catchments with high quality from? Please add this reference and also the reference for the inflow data of the multi-purpose dams to the reference list at the end of the manuscript.

-> We will provide the reference from the Ministry of Land, Transport and Maritime Affairs of the Korean government.

P3: Chapter on description of study area and data: where is the evaporation data from that you need as input for the runoff model? Do you need elevation data for the runoff model? Do you have any information about geology, vegetation etc. because you mention this as possible reason for poor top-kriging performance (P9 L15).

-> Evaporation is estimated from temperature data using the simple temperature-based model proposed by Oudin et al. (2005). Oudin et al. (2005) concluded that the scientific potential evaporation model was not good for daily runoff modeling with GR4J. They proposed a temperature-based model for GR4J together with the evaluation of numerous ET models. We will include this description in the methodology section. And, no elevation data are necessary for rainfall-runoff modeling. We will provide average slope, vegetation, urban areas of each catchment as recommended.

P4 L3-8: I recommend to add a short description of the structure of GR4J, information about the required input data and its resolution as well as information about the use of elevation bands.

-> We will add the description and information as advised.

P6 L24: Why do you evaluate the hydrograph with Nash-Sutcliffe and volume error, but the FDC only with Nash-Sutcliffe?

-> We will redo parameter calibrations with a criterion balanced between high and low

flows, e.g. a combination of NSE and log VE in Zhang et al. (2015), in revision. We will also apply this to FDC-based calibration for consistency. Then, we will separately evaluate performances with other measures (e.g. NSE, Log NSE, Pearson r and RMSE).

P6 L28: Give an explanation why you selected runoff ratio, baseflow index and rising limb density as signatures. Why three signatures?

-> They were regarded as major flow signatures in catchment classification (Sawicz et al., 2011). The runoff ratio explains average portion of precipitation that is discharged. Thus, it explains water holding capacity and evaporation loss of catchments. The baseflow index explains the portion of slow flows in hydrographs, and thus quick flows can also be evaluated together. Rising limb density shows how fast the catchment response is. Although there are more flow signatures (e.g., spectral density in hydrographs), we assumed the three signatures explain climatic, soil, and topographic characteristics of catchments. Snow day ratio is also an important signature, but we only focused on catchment response to liquid forcing, which is important in South Korea. We will add this point in the manuscript.

P7 L22: Please explain why you use 5 donor catchments and not 3 or 7.

-> It is for consistency with the TND interpolation using the top-kriging. We found five nearby catchments were best for interpolating TNDs. It is achieved from iterative calculations and explained in page 9. In revision, we explain n=5 in the section of FDC regionalization. We wanted to have consistency in the number of donor sets for both approaches for ungauged catchments. In addition, from results in a comparative study of Oudin et al. (2008), we were indicated that adding donor catchments would worsen predictive skills of proximity-based regionalization when using GR4J (see figure 6 in Oudin et al., 2008).

P8 L9: Again, why do you use 5 parameter sets and not 3 or 7? Does it make sense to give weights to these 5 parameter sets given the uncertainty related to them?

-> We preliminarily tested an average-sized catchment by plotting the number of parameter sets vs. NSEQ. We will add this explanation in revision. In fact, the weights did not sensitively affect our LOOCV because five parameter sets showed similar NSEFDC values for most catchments. We just hypothesized weighting parameter sets with higher NSEFDC would be better. As you commented, it would not be meaningful under given uncertainty. We will consider simple averaging in revision. P8 L31: Why do you use NSE(Q) 0.6 as threshold?

-> Oudin et al. (2008) discussed that low predictive performance at donor catchments clearly affected performance of ungauged catchments with GR4J. They used 0.7 of NSE for screening out catchments with low performance. In our study, 0.7 of NSE was too high to have adequate proximity between gauged and ungauged catchments. So, we reduced it to 0.6. However, in revision, we want to include all 44 catchments for regionalization irrespective of predictive performance in order to fully consider uncertainty sources in parameter regionalization.

P9 L4-10: This paragraph belongs to the methods section and should not be in the results. Do I understand correctly that 5 donors were used for the regionalization of the FDC? If yes, how do you get a total weight of 1? If no, please write this sentence more clearly.

-> We will move this sentence to the methodology section. It means that five donors were used for estimation of TNDs, which is an area between mean annual flow and below-mean flows in an FDC. The weights for estimating TNDs were used for regionalization of the FDCs too. Thus, the number of donor catchments for the TND interpolation is same as that for FDC regionalization. The sum of weights for TND interpolation is constrained as one when solving the ordinary kriging linear system, which is a part of top-kriging. Please find this information in Pugliese et al. (2014) or Skøien et al. (2006). We will explain this more clearly.

P9 L29: Why a threshold of NSE(FDC) of 0.8 is used? Is it necessary to reduce

the number of catchments used in model calibration by the additional constraint of NSE(FDC)? Wouldn't it be better to keep as many catchments as possible?

-> Because we only considered catchments with high performance in parameter regionalization, we need to apply the screening for calibration with regional FDCs. We agree that it would be better to consider all catchments irrespective of predictive performance. We will use all catchments in revision.

P9 L31: Figure 5b – how do you explain the scatter in the low flow?

-> This is because the plot is in log-scale. Thus, residuals in low flow appear significant. This plot is similar to Figure 8 in Pugliese et al. (2014). In our opinion, important metric is the NSE in the figure. For clarification, we will provide Log NSE together.

P12 L10: Discussion: The manuscript would strongly benefit from a deeper and more extensive discussion of the results with other studies. Many statements appear in the discussion for which it is not clear where they are taken from - so please cite other studies properly (e.g. first sentence in discussion). There is no chapter in the discussion about the prediction of the hydrograph in ungauged catchments.

-> We will provide references accordingly. As replied, we will provide in-depth discussion about hydrograph prediction in gauged and ungauged catchments when restructuring the manuscript.

P14 L1: Summary and conclusions: I recommend to shorten this chapter. Point 1 (L9-13) is in my opinion no key finding of the study, point 2 (L14-17) is more an assumption than a result and point 5 (L27-30) is also rather a hypothesis than a result and should be formulated as possible further steps.

-> We agreed. In revision, we will summarize and results accordingly. And, the evidence of additional signatures will be provided as mentioned.

P15: Please add the information on data availability and author contribution.

-> We add the data availability and author contributions as requested.

P20 L1: If you want to show the parameter ranges used by Perrin et al. (2003), you should also argue why you use different ones. Since there are only 4 parameters the information could also be added in the text.

-> We will revise the part as requested.

P24: Figure 4 – please add labels to the FDC-plot. Why did you select catchment 15 which is within the 50

-> We will revise the plots accordingly. It was a just random selection. We will do this more meaningfully in revision.

P26: Figure 6 – The authors often mention that the calibration with the regionalized FDC results in hydrographs with poor timing. Such timing issues are not obvious in the plots of Fig. 6. Thus, I would recommend to show time periods or catchments where timing really is a challenge.

-> We will redraw all plots in revision.

P27-29: Figures 7, 8 and 9: These plots all look very similar to me and I recommend to condense or reduce the information of these plots. In my opinion it is not necessary to show the calibration values, I would rather focus on validation efficiency because that's the tougher criteria. I also think it's not ideal to compare the regionalization approaches in this way and I recommend to use the concept of benchmarks for comparison: e.g. make the difference between the benchmark strategy (RFDC_cal) and the PROX_reg strategy. The use of benchmarks results in one single value which can easily be interpreted: e.g. positive values mean that PROX_reg is better than RFDC_cal.

-> As replied earlier. We will provide actual calibration results with an FDC and additional flow signatures. As recommended, we will take differences for improving readability.

[Figure]

P29: Figure 9: I like the idea of evaluating further signatures and using them as additional constraints in model calibration. However, this plot does not give enough evidence for the conclusions drawn. To show that RLD and RFDC_cal are uncorrelated different methods are needed. I am also not convinced that the additional use of RLD would improve model calibration with the regionalized FDC more than RQP.

-> We will provide actual calibration results with an FDC and additional flow signatures at gauged catchments.

Minor comments: Please use the HESS guidelines for all abbreviations, so that all are done in a similar style as e.g. the abbreviation of the baseflow index. Please also write Figure 9 and not Fig. 9 when you refer to it at the beginning of a sentence.

-> We will globally recheck all abbreviations and expressions.

P1 L1: I would adapt the title: ". . .and model calibration with regionalized flow-duration

-> We will rethink about the title in revision. The proposed the title will also be considered.

P1 L12: Shouldn't it say "Leave-one-out cross validation"?

-> Some studies used the term "cross-evaluation". If LOOCV is more familiar to readers, we will change it.

P3 L1: Why do you consider the selected signatures as "major signatures"?

-> Our response to this question is provided above.

P3 L11: Please provide numbers for the percentage of precipitation falling as snow.

-> We will provide them as requested.

P5 L25: Where does this equation for calculating MAP* come from? Why do you need the constant?

-> It is also provided by Pugliese et al. (2014). MAP* is just a multiplication of the

mean annual precipitation and the drainage area. Pugliese et al. (2014) was regarded MAP* as the mean annual flow for ungauged catchments. The constant is for a unit conversion from (mm yr-1 km2) to (m3 s-1).

P7 L14: I recommend to integrate this whole chapter in chapter 3.1, because it is about regionalization and not about evaluation.

-> We will restructure the manuscript, and consider this comment in revision.

P7 L19: This information is already in the introduction and is not needed in the methods part. Furthermore, you cite different studies here than in the introduction.

-> We will combine this part with the introduction.

P7 L25: I recommend to integrate this whole chapter in chapter 3.2, because it is about regionalization and not about evaluation.

-> We will restructure the manuscript, and consider this comment in revision.

P8 L2: I don't think that the regionalized FDC is used as objective function. It is rather used as reference value against which model simulations are evaluated.

-> We agreed. We will check expressions in the sentence.

P8 L25: Can you say what the CPI was for these catchments that were poorly modelled?

-> The CPI values will be provided in revision.

P10 L8: What about the efficiencies of catchment 13?

-> In revision, more comprehensive discussion will be provided. As replied, we need to recalibrate with a balanced criterion. Hydrograph simulations will be newly provided with improved discussions.

P10 L19: Please introduce the abbreviations such as PROX_reg earlier in the manuscript, because e.g. Fig. 5b already uses abbreviations.

[Figure]

-> We will consider this comment in revision.

P19 L1: Table caption should be adapted because NSE(Q) and NSE(FDC) are not catchment properties.

-> We will revise the caption as well.

P21: Figure caption - it's the catchments that are labeled in the center and not the numbers. Also skip ". . .for GR4J model and FDC regionalization"

-> We will revise the caption.

P23: Figure 3a – I agree that it is important to know that the model is able to simulate runoff in most catchments. However I don't think that a boxplot is needed for that. The median and the range of the model performance in calibration and validation could also just be mentioned in the text.

-> We agreed. We will remove it and comment on it in the text.

References

Blöschl, G., Sivapalan M., Wagener, T., Viglione, A., Savenije, H., 2013. Runoff Prediction in Ungauged Basins. Synthesis across Processes, Places, and Scales. Cambridge University Press. New York, USA. Hrachowitz, M. et al., 2013. A decade of Predictions in Ungauged Basins (PUB) - A review. Hydrolog. Sci. J., 58, 1198-1255, doi:10.1080/02626667.2013.803183.

Oudin, L., Andreassian, V., Perrin, C., Michel, C., Le Moine, N., 2008. Spatial proximity, physical similarity, regression and ungaged catchments: A comparison of regionalization approaches based on 913 French catchments, Water Resour. Res., 44, W03413, doi:10.1029/2007WR006240.

Oudin, L., Hervieu, F., Michel, C., Perrin, C., Andreassian, V., Anctil, F., Loumagne, C., 2005. Which potential evapotranspiration input for a lumped rainfall-runoff model? Part 2 – towards a simple and efficient potential evapotranspiration model for rainfall-runoff

modelling. J. Hydrol., 303, 290-306.

Pugliese, A., Castellarin, A., Brath, A., 2014. Geostatistical prediction of flow–duration curves in an index-flow framework, Hydrol. Earth Syst. Sci., 18, 3801-3816.

Pugliese, A., Farmer, W. H., Castellarin, A., Archfield, S. A., Vogel, R. M., 2016. Regional flow duration curves: Geostatistical techniques versus multivariate regression. Adv. Water Resour., 96, 11-22.

Sawicz, K., Wagener, T., Sivapalan, M., Troch, P.A., Carrillo, G., 2011. Catchment classification: empirical analysis of hydrologic similarity based on catchment function in the eastern USA. Hydrol. Earth Syst. Sci., 15, 2895-2911.

Shu, C., Ouarda, T.B.M.J, 2012. Improved methods for daily streamflow estimates at ungauged sites. Water Resour. Res. 48, W02523, doi:10.1029/2011WR011501.

Skøien, J. O., Merz, R., Blöschl, G., 2006. Top-kriging – geostatistics on stream networks. Hydro. Earth Syst. Sci., 10, 277-287.

Walter, M. T., Brooks, E.S., McCool, D.K., King, L.G., Molnau, M., and Boll, J., 2005. Process-based snowmelt modeling: does it require more input data than temperature-index modeling?. J. Hydrol., 300, 65-75.

Zhang, Y., Vaze J., Chiew, F.H.S., Li, M., 2015. Comparing flow duration curve and rainfall-runoff modelling for predicting daily runoff in ungauged catchments. J. Hydrol., 525, 72-86.
* * *

---

## Author Comment (AC2) · 23 Dec 2016

Dear anonymous referee:

We greatly appreciate your valuable contribution to our manuscript, and thank for your favorable comments. Accepting referee 3's constructive comments, we will improve the manuscript in revision, and your comments will be considered together. We will restructure the manuscript as:

(1) We will include more literatures about prediction in ungauged catchments in introduction. Additional literatures will be about FDC regionalization and signature-based calibration methods. We will also introduce the decade-long project of the IAHS in

Prediction in Ungauged Basins (Blöschl et al., 2013; Hrachowitz et al., 2013).

(2) We will restructure the manuscript from gauged to ungauged catchments. We will first show results and discussion about predictive performance and uncertainty at gauged catchment of both hydrograph- and FDC-based calibrations. Then, we will move to compare and discuss the parameter regionalization and calibration with regional FDCs.

(3) We will provide actual FDC-based calibration in combination with three flow signatures at gauged catchments. This will provide interest to readers. Once again, we thank for your contribution, and please find our response as per your comment below.

1. The authors rely often on to NSE efficiencies to assess the reliability of each model, and they conclude that, given the good results obtained with PROXreg, which is the regionalization of the parameters of the rainfall runoff model, that this model is preferable to the other given its capacity to reproduce the true idrograph in time. However, this conclusion seems to strongly contrast with other performance indices they deliver, e.g. VEQ, RQP, IBF, so I would soften a little the conclusions and would let the judgment be more flexible.

-> Accepting constructive comments from the referee 3, we will restructure our manuscript in revision. We will discuss hydrograph-based and FDC-based calibrations at gauged catchments first, and then move to the ungauged case (i.e. parameter regionalization and FDC regionalization). And, we will provide actual calibration results with combination between FDCs and other signatures. Hence, new discussion and conclusions will be provided in revision.

2. NSE is the most used performance index in literature and I agree with using it, but recently it has been criticized its capacity to understand how a model produce good result with low flows, while it emphasizes the capacity to understand high flows, so that NSE could end to be a biased index (the authors also recall this behavior at P11 L8). I wonder how the final performances vary by adopting LNSE, which is the same as

NSE but taking the natural logs of streamflows. I assume that the lesser influence of the low flow regime prediction into NSE might somehow introduce a distortion of final judgment. In case the results are substantially different I would recommend the authors to make an effort in discussing these results as well.

-> We agreed. In revision, we will use a balanced criterion between high and low flows for calibrations (e.g. Zhang et al., 2015). Evaluation will be done for both high and low flows too (e.g. using NSE and log NSE).

3. I would suggest to change the title including the word "streamflow" somewhere. In this way, the topic is clearer. Perhaps "A comparison between parameter regionalization and model calibration with flow duration curves for prediction of streamflows in ungauged catchments" ?

-> The manuscript will be retitled after revision. We will consider this comment when retitling.

4. Table 1 reports for each model two columns in which the author say whether or not NSE efficiencies are greater than 0.6 or 0.8 respectively. I think that would be of more interest for the reader to see all the values for each catchment as well as the cut-off at 0.6 or 0.8. For instance, they can report efficiencies for each catchments and let the numbers above the cutoff in bold face. Reporting "Y" or"N" only might result uninformative and, at the very end, useless.

-> In revision, we will use all catchments regardless of performance measures for having more proximity between gauged and ungauged catchments. The table will be updated with new performance measure as you recommended.

5. I would recommend to extend paragraphs 3.3.1 and 3.3.2 to introduce some more details of the two proposed approaches. Furthermore, I would move those two paragraphs before the evaluation indices adopted.

-> This part will be moved to introduction. We will provide more details about two
approaches in revision.

Technical notes and misspellings

1. P3 L7. "Siberian high pressure", perhaps is "low pressure".

-> "high pressure" is correct because we intended to explain cold and dry weather conditions in winter seasons in South Korea.

2. P8 L19. Please add the word "between" in between the words "coefficients" and "CPI".

-> The manuscript will be restructured. We will globally check grammatical errors in revision.

3. P10 L3. Please remove the article "the" between the words "values" and "between".

-> We will globally check technical and grammatical errors in revision.

4. P10 L4. "Based on the high [. . .]" is perhaps "Based on the highest [. . .]"?.

-> We will globally check technical and grammatical errors in revision.

5. I have not completely understood what the author mean with "orthogonal" referring to streamflow signatures, please consider to add some more details arguing what does this word mean for them into the context of the sentence.

-> In our study, an "orthogonal" signature is one that can supplement the FDC for parameter calibration independently. Hrachowitz et al.(2013) used this expression for hydrologic observations that seem to be independent each other, albeit not all variables are strictly independent. We will more clearly define this term in revision.

References

Blöschl, G., Sivapalan M., Wagener, T., Viglione, A., Savenije, H., 2013. Runoff Prediction in Ungauged Basins. Synthesis across Processes, Places, and Scales. Cambridge University Press. New York, USA.

[Figure]

Hrachowitz, M. et al., 2013. A decade of Predictions in Ungauged Basins (PUB)—a review, Hydrolog. Sci. J., 58, 1198-1255, doi:10.1080/02626667.2013.803183.

Zhang, Y., Vaze J., Chiew, F.H.S., Li, M., 2015. Comparing flow duration curve and rainfall-runoff modelling for predicting daily runoff in ungauged catchments. J. Hydrol., 525, 72-86.
* * *

---

## Author Comment (AC3) · 23 Dec 2016

Dear Dr. Qamar:

We greatly appreciate your interest and comments in our manuscript. We believe that revising our manuscript based on the referee 3's constructive review will improve the manuscript. Your comments will be considered as well in revision. Our main revision will mainly include follows points:

(1) We will include more literatures about prediction in ungauged catchments in introduction. Additional literatures will be about FDC regionalization and signature-based calibration methods. We will also introduce the decade-long project of the IAHS in

[Figure]

Prediction in Ungauged Basins (Blöschl et al., 2014).

(2) We will restructure the manuscript from gauged to ungauged catchments. We will first show results and discussion about predictive performance and uncertainty at gauged catchment of both hydrograph- and FDC-based calibrations. Then, we will move to compare and discuss the parameter regionalization and calibration with regional FDCs.

(3) We will provide actual FDC-based calibration in combination with three flow signatures at gauged catchments. This will be more interesting to readers.

Once again, we thank for your contribution, and please find our response as per your comment below.

The manuscript "A comparison between parameter regionalization and model calibration with flow duration curves for prediction in ungauged catchments" compares parameter regionalization techniques with FDC-based model calibration. My specific comments are listed below;

1. A number of studies have already been conducted regarding the comparison parametric and non-parametric methods for the regionalization of FDCs. Some of the studies are listed below; Ganora D, Claps P, Laio F, Viglione A. 2009. An approach to estimate non-parametric flow duration curves in ungauged basins. Water Resour Res. 45. doi:10.1029/2008WR007472 Qamar MU, Azmat M, Cheema MJM, Shahid MA, Khushnood RA, Ahmad S (2016) Model swapping: a comparative performance signature for the prediction of flow duration curves in ungauged basins. J Hydrol 541:1030–1041. http://dx.doi.org/10.1016/j.jhydrol.2016.08.012 The authors are advised to go through these studies in order to familiarize themselves with the latest developments in the field of PUB. They further have to defend how their study is different from the already executed comparative studies? Frankly, I don't see much innovation, here.

-> We will provide more studies on FDC regionalization and signature-based calibration as recommended by the referee 3. We would consider the proposed references in revision. In our knowledge, the FDC-based model calibration has been barely evaluated against conventional parameter regionalization. Given numerous methods for runoff prediction in ungauged catchments, comparative studies can significantly contribute to selecting proper methods for hydrologic applications (e.g. Zhang et al., 2015; Parajka et al., 2013) in our opinion. The objective of this study is not to propose a novel approach.

2. The authors used dataset from 2007-2015 for model calibration and validation phases. Such dataset is not enough to hunch the flow trends. In such case, the modeling technique can be considered suitable for a particular time phase but cannot be generalized due to inadequate data length. Since the results generated by the proposed model for the entire study area are tested by using a LOOCV procedure. One solution to increase the data length is to consider one station as ungauged, removing it from the whole database and estimating FDC for that station with the proposed approach.

-> As explained, quality of streamflow data before 2007 was a critical reason for only using a relatively short period. In fact, data-length for calibration is a controversial topic. Several studies (e.g. Seibert and Beven, 2009) indicated that a few runoff measurements can contain much of the information content of continuous runoff time series. Since droughts and floods were all experienced in South Korea during 2011-2015, we could assume the calibrated parameter sets can relatively well reflect the hydrologic responses. For your information, we provide here the time series plots of spatially averaged SPI6, SPEI6, and SEDI6 during 1974-2015 in South Korea. SEDI is a recently proposed drought index based only on evapotranspiration (Kim and Rhee, 2016) And, I do not clearly understand why leaving out one catchment can extend the data length. The LOOCV is to evaluate performance of methods in the ungauged cases.

3. Line 15 reads, "Though combining a temperature index snowmelt model with GR4J can be an alternative approach, it increases the number of parameters and thus model

uncertainty". How can increase of parameters make the technique uncertain? Increasing parameters increase complexity but it always betters efficiency. A thorough explanation is needed for this claim.

-> GR4J conceptualizes catchment response to rainfall using 4 parameters. If we add any parameter for snowmelt process, it can affect the existing parameters. Interactions between the parameter of snowmelt and the other parameters can happen when calibrating against a hydrograph. Thus, uncertainty to determine parameters will be increased, and the equi-finality problem will become severe.

4. The periods and range of streamflow data (2007-2015) and climatic data (1973-2015) are not overlapping? Will it not be problematic? Moreover how was the range of climatic data used in calibration and validation phases of the model?

-> They are overlapping during the period of streamflow data (2007-2015). I do not understand intention of this comment exactly. As explained, a two-year warm-up period is used. For example, when simulating runoff for 2011-2015, we simulated streamflow for 2009-2015, but evaluated for 2011-2015 only.

5. The equitation (4) seems to be generated by multivariate regression analysis. The authors never explained its generation, which is inevitable. How effective was this rescaling?

-> No. It is just a multiplication of annual precipitation and drainage area. Annual precipitation (mm yr-1) is rescaled by the drainage area (km2) for having a unit of streamflow (m3 s-1).

6. Equation 6 should be eliminated as it is already discussed above (equation 2). Moreover the performances indices need to increased. I suggest including mean absolute error and root mean square error.

-> We will consider unifying similar criteria in revision. Strictly speaking, equation 2 and 6 are different. The performance indicators will be reselected based on previous

studies in revision.

7. Why only five nearest neighbors were used? Why not, say, 8 or 10 or 12? There has to be a reason for that. The authors are suggested to go through the following paper and study Figure 9 in detail in which Samaniego and Kumar (2010) selected nearest neighbors by observing the error generated by different number of neighbors.

Samaniego, L., A. Bárdossy, and R. Kumar (2010), Streamflow prediction in ungauged catchments using copula-based dissimilarity measures, Water Resour. Res., 46, W02506, doi:10.1029/2008WR007695.

-> This was because GR4J showed rapidly decreasing performance with increasing neighbors in Oudin et al. (2008). Five catchments are for consistency with FDC regionalization. We will address this more clearly in revision.

8. The authors never discussed the complications involved in the implementation of each method. The discussion section should also compare the simplicity of each method in terms of implementation.

-> As replied, we will restructure the manuscript, and provide more comprehensive discussion in revision.

References

Kim, D., Rhee, J., 2016. A drought index based on actual evapotranspiration from the Bouchet hypothesis. Geophys. Res. Lett., 43, 10,277–10,285, doi:10.1002/2016GL070302.

Oudin, L., Andreassian, V., Perrin, C., Michel, C., Le Moine, N., 2008. Spatial proximity, physical similarity, regression and ungaged catchments: A comparison of regionalization approaches based on 913 French catchments, Water Resour. Res., 44, W03413, doi:10.1029/2007WR006240.

Parajka, J., Viglione, A., Rogger, M., Salinas, J.L., Sivapalan, M., Bloshl G., 2013.

Comparative assessment of predictions in ungauged catchment – part 1: Runoff-hydrograph studies. Hydrol. Earth Syst. Sci., 17, 1783-1795.

Seibert, J., Beven, K.J., 2009. Gauging the ungauged basins: how many discharge measurements are needed?. Hydrol. Earth Syst. Sci., 13, 883-892.

Zhang, Y., Vaze J., Chiew, F.H.S., Li, M., 2015. Comparing flow duration curve and rainfall-runoff modelling for predicting daily runoff in ungauged catchments. J. Hydrol., 525, 72-86.

[Figure]

**Fig. 1.** Drought indices during 1974-2015 in South Korea

---

## Author Comment (AC4) · 23 Dec 2016

Dear anonymous referee:

We greatly appreciate your valuable contribution to our manuscript, and thank for your comments. Accepting referee 3's constructive comments, we want to improve the manuscript. Your comments will be considered together in revision. Our main direction for restructuring the manuscript is:

(1) We will include more literatures about prediction in ungauged catchments in introduction. Additional literatures will be provided about FDC regionalization and signature-based calibration methods. We will also introduce the decade-long project

of the IAHS in Prediction in Ungauged Basins (Blöschl et al., 2013; Hrachowitz et al., 2013).

(2) We will restructure the manuscript from gauged to ungauged catchments. First, we will show results and discussion about predictive performance and uncertainty at gauged catchment of both hydrograph- and FDC-based calibrations. Then, we will move to compare and discuss the parameter regionalization and calibration with regional FDCs.

(3) We will provide actual FDC-based calibration in combination with three flow signatures at gauged catchments. This will provide interest to readers.

Once again, we are very thankful for your contribution, and please find our response as per your comment below.

This study compares two regionalisation methods for runoff predictions: the traditional spatial proximity and the model calibration against regionalised flow duration curves. I found this study has a very limited contribution to the predictions in ungauged basins (PUB). The major reasons are as follows: 1. There are numerous studies carried out by using various methods for PUB. The hydrological modelling uses spatial proximity, physical similarity and regression to regionalize its calibrated parameter sets to un-gauged catchments. There are several studies that used flow duration curve methods for runoff predictions (Shu and Ouarda, 2012; Zhang et al., 2015), which just use ob-served runoff and catchment attributes to predict runoff time series, but does not need to involve any hydrological modelling. This kind of research is totally ignored by the authors. These researches use the three steps to predict daily runoff: (1) building FDC method (geostatistical methods, statistical methods, etc); (2) estimating flow quantile based on some assumptions; (3) predicting runoff time series. The predictions results are very impressive (see Zhang et al., 2015).

-> We disagree that our approach has limited contribution. A FDC method can show impressive performance; however, it does not guarantee that the FDC method is always

the best for runoff prediction. The existence of good FDC methods is not relevant to our study. We are focusing on rainfall-runoff modelling in ungauged catchments. Indeed, we never stated that the proximity-based parameter regionalization is the best approach for runoff prediction in ungauged catchments in general. In addition, FDC methods are of relative merits and shortcomings. Despite simplicity of FDC methods, they could be useless when no streamflow data available around the target catchments, except ones using duration of precipitation (e.g. Smakhtin and Masse, 2000; Kim and Kaluarachchi, 2014). In our opinion, streamflow data are generally less available than climatic forcing data (i.e. precipitation and temperatures). The objective of our study is not to directly compare between a FDC method and a parameter regionalization as did Zhang et al. (2015). The comparative study of Zhang et al. (2015) is fundamentally different from our study. We are discussing how to have parameter sets in ungauged catchments when using a frequently used conceptual model for runoff simulations. The two approaches in our study are about how to generate streamflow from atmospheric forcing. The destination of a FDC method and a rainfall runoff model is same (i.e. daily runoff estimation), but the FDC method is to transfer streamflow data at gauged to ungauged catchment. We never intended to directly transfer observed streamflow to ungauged catchments.

As we introduced, parameter calibration with regional FDCs for rainfall-runoff modeling is barely evaluated against conventional parameter regionalization. Both approaches are to have parameter sets for a rainfall-runoff model in ungauged catchments. Can we assure one is better than the other? There is no calibration process in parameter regionalization for ungauged catchments while the FDC-based calibration has it. We intended to answer this question. As you commented, there are a number of methods for runoff prediction in ungauged catchment. This fact rather makes comparative assessments (e.g., Zhang et al., 2015, Parajka et al., 2013, and our study) very important. We believe that a comparative assessment contributes to making a better selection among various methods.

2. It is not surprise at all to see the calibration against regionalised flow duration curves performs worse than the traditional spatial proximity approach for runoff time series predictions since it does not include any runoff timing information, which is the key for runoff time series predictions.

-> We disagree that the calibration with regionalized FDCs is expected to be worse than the parameter regionalization. Regionalized parameter sets are not ones calibrated to runoff time series at outlets of ungauged catchments. Even though the parameters are calibrated to observed hydrographs at gauged catchments, they have additional uncertainty sources because of regionalization. If regionalized parameters have significant uncertainty, the calibration against regionalized FDCs could be better.

3. The sample number used here is too small. There are only 45 catchments used to evaluate regionalisation skill. Therefore, it is hard to get a generalised conclusion. Moreover, the authors only picked up 28 with good calibration of GR4J and FDC, making the sampling number extremely small.

-> We did not intend to provide general conclusions, but a case comparative study in South Korea. The number of gauged catchments is a given condition in South Korea. Indeed, some regionalization studies have less numbers of gauged catchments (see Figure 4 in Parajka et al., 2013). We believe that "extremely small" is overstating. In revision, we will use all 45 catchments for LOOCV to investigate overall uncertainty in the parameter regionalization and the FDC-based calibration.

4. The objective function. The selection of objective functions has very important implication on the conclusions. The authors used the classic NSE for hydrological modelling calibration/regionalisation. It will be inevitable that the predictions from spatial proximity regionalisation are better for high flow, but poorer for low flow. For a comprehensive evaluation, an objective function that compromises high flow and low flow (i.e. Box-Cox transformed streamflow) should be used.

-> We agreed. In revision, we will use a calibration criterion balanced between high

and low flows as in Zhang et al. (2015). And, runoff simulations will be evaluated for high and low flows (e.g. using NSE and log NSE)

Specific comments:

1. Introduction. It is not comprehensive. Lots of methods used for building FDC in ungauged catchments are not introduced. Lots of studies using FDC to predict runoff time series are ignored (see the above-mentioned are just some examples). Lots of spatial proximity regionalisation studies are not included. The authors should have a comprehensive literature review from ISI Web of Knowledge.

-> In revision, we will provide more studies on signature-based calibration and FDC regionalization. We will make the introduction more comprehensive.

2. It is very confused for the streamflow gauges used. The authors state that the 45 streamflow gauges used in this study are with negligible regulations (river diversion and dam operation), but they also state that "... operationally recorded at 16 multipurpose dams for the Water Resources Management Information ..." Is it really all the gauges are with negligible regulation?

-> In revision, we will correct the expression "operationally". It means inflows of the dams are routinely monitored for operating reservoirs for areas below the dams. Outflows are regulated, but inflows are limitedly altered. In our sense, all gauges showed natural flow regimes based on the correlation coefficient between CPI and runoff data. If human alteration is significant, we should have outliers from the correlation coefficients

3. Are all gauges not nested? Please clarify

-> Some of them are nested. For clarification, we will place catchment numbers at outlet locations in Figure 1.

4. Cross-validation and regionalisation. I am not against the cross validation (2011-2015) for model calibration and 2007-2010 for model cross validation). For regionalisation, I suggest to use the full period of dataset. Bury in mind, there are only nine years data for each gauge.

-> We will consider the entire data period for regionalization. However, we can do nothing about data length. As explained, runoff data before 2007 have poor quality. The influence of data lengths on model calibration is a controversial topic (Seibert and Beven, 2009). Sometimes data quality is more important. During 2009-2015, Korea had significant variation in climatic conditions as shown in the time series of drought indices (Fig. 1 in this response). Thus, we could hypothesize that runoff data have adequate information for calibration though long-term topographic changes cannot be considered.

5. GR4J requires precipitation and potential evapotranspiration for model inputs. It is not clear how the potential evaporation is calculated.

-> We used the temperature-based model proposed by Oudin et al. (2005) for GR4J. We will explain this in revision.

6. Objective function. To have a comprehensive evaluation of these two methods, please also include a Box-Cox transformed streamflow objective function.

-> As replied earlier, we will use a calibration criterion that can consider both high and low flows in revision.

7. First paragraph in section 3.3.1. Please include more references for the three regionalisation approaches.

-> We will provide more related studies in revision

8. Equation (4). How is the constant 3.171 * 10Ë Ę-5 derived?

-> This is a simple unit conversion from (mm yr-1 km2) to (m3 s-1). (10-3 m)×(86400×365 sec)-1×(106 m2) = 3.171×10-5 (m3 s-1).

9. The methods mixed with results. Half of section 4.2 should be moved to methodology

-> We will move the sentences to the methodology section.

10. Use all gauges for regionalisation. Please use all 45 gauges for the regionalisation. It makes no senses to me to exclude the 17 catchments with low NSEFDC (<0.80). You can setup prerequisite for PUB. It is not fair for another approach.

-> We will use all 45 catchments for two approaches as replied.

11. Figure captions. It is hard to follow figure captions. Please spell out all the abbreviations. I spent lots of time to figure out these abbreviations.

-> We will recheck the captions in revision.

References

Blöschl, G., Sivapalan M., Wagener, T., Viglione, A., Savenije, H., 2013. Runoff Prediction in Ungauged Basins. Synthesis across Processes, Places, and Scales. Cambridge University Press. New York, USA. Hrachowitz, M. et al., 2013. A decade of Predictions in Ungauged Basins (PUB) - A review. Hydrolog. Sci. J., 58, 1198-1255, doi:10.1080/02626667.2013.803183.

Kim, D. and Kaluarachchi, J., 2014. Predicting streamflows in snowmelt-driven watersheds using the flow duration curve method. Hydrol. Earth Syst. Sci., 18, 1679–1693.

Oudin, L., Hervieu, F., Michel, C., Perrin, C., Andreassian, V., Anctil, F., Loumagne, C., 2005. Which potential evapotranspiration input for a lumped rainfall-runoff model? Part 2 – towards a simple and efficient potential evapotranspiration model for rainfall-runoff modelling. J. Hydrol., 303, 290-306.

Parajka, J., Viglione, A., Rogger, M., Salinas, J.L., Sivapalan, M., Bloshl G., 2013. Comparative assessment of predictions in ungauged catchment – part 1: Runoff-hydrograph studies. Hydrol. Earth Syst. Sci., 17, 1783-1795.

Smakhtin, V.Y., Masse, B., 2000. Continuous daily hydrograph simulation using duration curves of a precipitation index, Hydrol. Process., 14, 1083–1100.

Zhang, Y., Vaze J., Chiew, F.H.S., Li, M., 2015. Comparing flow duration curve and rainfall-runoff modelling for predicting daily runoff in ungauged catchments. J. Hydrol., 525, 72-86.

———————————————

[Figure]

[Figure]

**Fig. 1.** Drought indices during 1974-2015 in South Korea